EMBO
Molecular Medicine

# Inhibition of DDR1-BCR signalling by nilotinib as a new therapeutic strategy for metastatic colorectal cancer

Maya Jeitany[1,†], Cédric Leroy[1,2,3,†], Priscillia Tosti[1,†], Marie Lafitte[1], Jordy Le Guet[1], Valérie Simon[1], Debora Bonenfant[2], Bruno Robert[4], Fanny Grillet[5], Caroline Mollevi[4], Safia El Messaoudi[4], Amaëlle Otandault[4], Lucile Canterel-Thouennon[4], Muriel Busson[4], Alain R Thierry[4], Pierre Martineau[4], Julie Pannequin[5], Serge Roche[1,*,†] (iD) & Audrey Sirvent[1,†,**] (iD)

## Abstract

The clinical management of metastatic colorectal cancer (mCRC) faces major challenges. Here, we show that nilotinib, a clinically approved drug for chronic myeloid leukaemia, strongly inhibits human CRC cell invasion *in vitro* and reduces their metastatic potential in intrasplenic tumour mouse models. Nilotinib acts by inhibiting the kinase activity of DDR1, a receptor tyrosine kinase for collagens, which we identified as a RAS-independent inducer of CRC metastasis. Using quantitative phosphoproteomics, we identified BCR as a new DDR1 substrate and demonstrated that nilotinib prevents DDR1-mediated BCR phosphorylation on Tyr177, which is important for maintaining β-catenin transcriptional activity necessary for tumour cell invasion. DDR1 kinase inhibition also reduced the invasion of patient-derived metastatic and circulating CRC cell lines. Collectively, our results indicate that the targeting DDR1 kinase activity with nilotinib may be beneficial for patients with mCRC.

**Keywords** collagen receptor; colorectal cancer; invasion; targeted therapy; tyrosine kinase
**Subject Categories** Cancer; Digestive System; Pharmacology & Drug Discovery

## Introduction

Colorectal cancer (CRC) is one of the leading causes of malignancy-related death worldwide. Most of these cancers are sporadic and under the control of genetic, epigenetic and environmental factors. The current clinical management involves surgical removal of the primary tumour, often associated with chemotherapy. However, tumour recurrence occurs in about 50% of patients at a distant site, resulting in bad prognosis and a 10% of survival at 5 years. Therapeutic failure is often associated with metastatic spread, in which cancer cells escape the primary tumour to disseminate in the circulation and establish secondary lesions in distant organs, mainly the liver (Dienstmann *et al*, 2017). Metastatic cell behaviour is characterized by the invasive properties and tumour-initiating capacity of disseminated CRC cells (Vanharanta & Massague, 2013). Currently, much research is focused on the signalling pathways that promote these metastatic properties and tyrosine kinases (TK) have emerged as important determinants of this process (Sirvent *et al*, 2012; Vanharanta & Massague, 2013). For instance, anti-epidermal growth factor receptor (EGFR) antibodies significantly improve the survival of patients with metastatic CRC (mCRC) (Lievre *et al*, 2006, 2008); however, only patients with wild-type RAS CRC (40% of cases) may hope to benefit from this treatment. Indeed, many patients harbour oncogenic mutations of RAS signalling components, and thus, the RAS cascade may be constitutively activated, independently from EGFR. In addition, prolonged EGFR-blockade induces tumour resistance that is often associated with RAS pathway reactivation (Misale *et al*, 2014). Therefore, there is a need for RAS-independent therapeutic strategies to reduce CRC resistance and metastatic progression.

Clinically approved drugs designed to target a well-defined oncogene may be of broader clinical interest through off-target-dependent mechanisms. For instance, nilotinib belongs to therapeutic agents used in the clinic that inhibit the tyrosine kinase activity of the fusion oncogene BCR-ABL and are highly effective against Chronic Myeloid Leukaemia (CML). The small-molecule imatinib (an ABL inhibitor) is the common treatment for chronic phase CML

1 CRBM, CNRS, University Montpellier, Montpellier, France
2 Novartis Institutes for Biomedical Research, Postfach, Basel, Switzerland
3 Actelion Pharmaceuticals Ltd, Allschwil, Switzerland
4 IRCM, INSERM, University Montpellier, Montpellier, France
5 IGF, CNRS, INSERM, University Montpellier, Montpellier, France
  *Corresponding author. Tel: +33 434359520; E-mail: serge.roche@crbm.cnrs.fr
  **Corresponding author. Tel: +33 434359503; E-mail: audrey.sirvent@crbm.cnrs.fr
  †These authors contributed equally to this work

(Druker et al, 2001); however, resistances emerge mainly caused by point mutations in the kinase domain of BCR-ABL that lower the inhibitor affinity. Therefore, nilotinib has been developed and approved for the treatment of patients with resistance to imatinib, with the notable exception of the BCR-ABL T315I mutant. Interestingly, nilotinib profiling by chemical proteomic methods led to the discovery that DDR1 is the highest affinity target of nilotinib in CML (Rix et al, 2007). DDR1 is a poorly characterized RTK that binds to collagens, ones of the major components of the extracellular matrix, and that functions as a central extracellular matrix sensor to regulate cell adhesion (Vogel et al, 1997). DDR1 also cross-talks with several transmembrane receptors, including Notch and TGF-β receptors, and could influence their signalling upon collagen stimulation (Leitinger, 2014). DDR1 may also promote cell proliferation, motility and invasion, depending on the tumour type and the nature of the microenvironment (Leitinger, 2014; Rammal et al, 2016). Curiously, the tumour role of DDR1 kinase activity is poorly documented. This activity seems to be dispensable for most DDR1 functions reported in human cancers, such as collective cell migration of squamous cell carcinoma (Hidalgo-Carcedo et al, 2011), breast tumour cell invasion (Juin et al, 2014) and metastatic reactivation in breast cancer (Gao et al, 2016). Moreover, very little is known about the mechanisms underlying DDR1 kinase activation and signalling, which is unusually slow and sustained over time (Vogel et al, 1997). One notable exception is lung cancer where KRAS mutations induce DDR1 expression to sustain Notch oncogenic signalling and tumorigenesis. DDR1 pharmacological inhibition reduces lung tumour progression in mouse and patient-derived xenograft models, suggesting an important kinase-dependent function of DDR1 in this cancer (Ambrogio et al, 2016). However, the mechanism by which DDR1 kinase activity modulates Notch signalling is currently unknown.

Here, we report an additional important DDR1 kinase-dependent function in CRC metastasis formation. We show that DDR1 pharmacological inhibition by nilotinib inhibits the metastatic behaviour of CRC cells through a RAS-independent mechanism. By phosphoproteomic analysis, we then identified BCR as an important DDR1 substrate involved in the maintenance of the β-catenin oncogenic signalling necessary for tumour cell invasion. We then showed that nilotinib can inhibit the DDR1-mediated invasive potential of CRC cells in a liver metastasis mouse model and of patient-derived cell lines originating from metastatic tumours and circulating CRC cells. Collectively, our results indicate that targeting DDR1 kinase activity with nilotinib could be of therapeutic interest for patients with advanced CRC.

# Results

### Nilotinib displays anti-metastatic activity in CRC

We found that the clinically approved TKI nilotinib (100 nM) inhibits invasive activity of a panel of CRC cell lines in Boyden chamber assays, irrespective of their KRAS/BRAF status (Fig 1A). Similar results were obtained in 3D spheroid assays where the HCT116 CRC cells were embedded in collagen I matrix (Fig 1B). The deduced $IC_{50}$ was in the 15–30 nM range in HCT116 cells (Fig 1C) and was comparable to nilotinib growth inhibitory potency in leukaemic cells

(Weisberg et al, 2005). We then evaluated nilotinib activity in vivo using a CRC liver metastasis model in nude mice, in which HCT116 cells injected in the spleen of recipient animals colonize the liver via the hepatic portal vein. A daily regimen of 50 mg/kg nilotinib, a dose that shows anti-leukaemic activity in experimental models (Weisberg et al, 2005), strongly inhibited liver metastasis formation in treated animals compared with untreated controls (vehicle) (Fig 1D). Thus, nilotinib can inhibit CRC cell invasion and metastasis formation.

### DDR1 promotes CRC metastasis formation

We then analysed the molecular mechanism underlying nilotinib anti-tumour activity in CRC. As ABL is not mutated in this cancer, we speculated the involvement of DDR1, the other major target of this inhibitor that was identified using a chemical proteomic approach (Rix et al, 2007). To test this hypothesis, we first assessed DDR1 role in CRC metastasis formation. DDR1 was well expressed and active in all tested CRC cell lines, as shown by the increased tyrosine phosphorylation upon collagen I stimulation (Fig EV1A). Moreover, DDR1 depletion with two different shRNAs strongly inhibited the invasive capacities of KRAS-mutated HCT116 and BRAF-mutated HT29 CRC cells in vitro (Fig 2A–C). DDR1 silencing also significantly reduced the metastatic potential of HCT116 cells in vivo after intrasplenic inoculation in nude mice (Fig 2D). This anti-tumour effect was confirmed by the large decrease in circulating tumour DNA (ctDNA) level that was used as biomarker of metastasis formation in these animals (Mouliere et al, 2013; Thierry et al, 2014; Fig 2D). Conversely, DDR1 overexpression in HT29 and in KRAS-mutated SW620 CRC cells significantly increased their invasive properties in vitro (Fig 2E and F) as well as liver metastasis development in vivo in nude mice upon intrasplenic inoculation of DDR1-overexpressing SW620 cells (SW620-DDR1 cells) compared with controls (mock transfected) (Fig 2G). In agreement, ctDNA level also was increased in inoculated animals compared with controls (mock) (Fig 2G). These results confirmed DDR1 role in CRC metastasis formation.

### Nilotinib anti-tumour activity is mediated by DDR1 kinase inhibition

We next determined the role of DDR1 kinase activity in nilotinib anti-tumour activities in CRC. We found that increasing doses of nilotinib (0.8–100 nM) inhibited the slow, but persistent increase in DDR1 tyrosine phosphorylation induced by collagen I (Fig EV1B). The deduced $IC_{50}$ was in the 1–8 nM range, which tightly correlated with its anti-invasive activity in these tumour cells. Surprisingly, addition of nilotinib for only 15 min was enough to abrogate DDR1 tyrosine phosphorylation induced by overnight collagen I stimulation (Fig EV1C), suggesting that DDR1 activity is tightly regulated by phosphatases in CRC cells.

Nilotinib resistance in leukaemia is often caused by the mutation of threonine 315 in BCR-ABL into a bulkier isoleucine that reduces the drug affinity for this kinase (Weisberg et al, 2007). This mechanism appeared to be conserved in DDR1 TK domain because the corresponding mutation (T701I) reduced nilotinib inhibition of collagen I-induced tyrosine phosphorylation of mutant DDR1 by 50-fold compared with wild-type DDR1 (WT) (Fig 3A). We then used this mutant to assess DDR1 role in nilotinib anti-tumour activity in

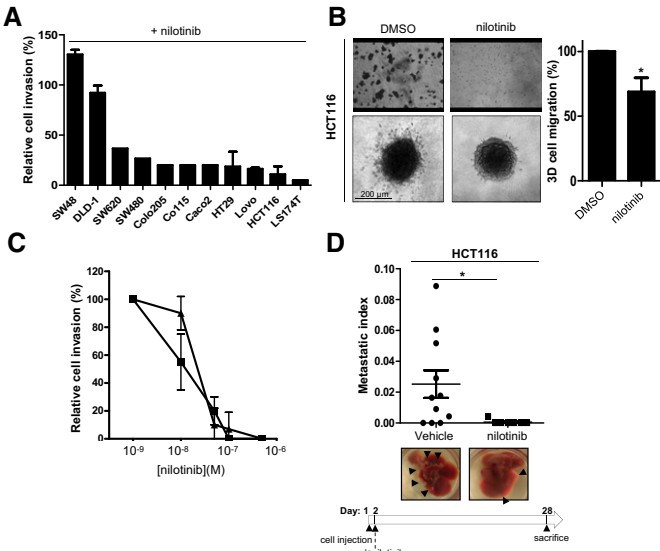

**Figure 1.  Nilotinib has anti-metastatic activity in CRC.**

A   Nilotinib anti-invasive activity in a panel of CRC cell lines. Percentage of cell inhibition in Boyden chambers containing 1 mg/ml Matrigel for the indicated CRC cell lines incubated with 100 nM nilotinib (mean ± SEM; *n* = 3).

B   Nilotinib activity in Boyden chambers and spheroid assays. HCT116 cells incubated with 100 nM nilotinib or DMSO (control) were seeded in the upper compartment of a Boyden chamber containing 1 mg/ml Matrigel for 24 h or embedded as spheroids in collagen I matrix for 72 h and then imaged by phase-contrast microscopy. The histogram shows the percentage of migrating cells in the collagen I matrix normalized to control condition set at 100% (mean ± SEM; *n* = 3 independent experiments with six replicates; *P < 0.05 Student's *t*-test). The quantification of nilotinib anti-invasive activity in HCT116 cells is shown in panel (C).

C   Dose–response curve of nilotinib effect on CRC cell invasion. Percentage of cell invasion (relative to control) in Boyden chambers containing 1 mg/ml Matrigel of HT29 (▲) and HCT116 (■) CRC cells treated with the indicated concentrations of nilotinib (mean ± SEM; *n* = 3).

D   Nilotinib anti-metastatic activity in nude mice. HCT116 cells were injected in the spleen of nude mice (*n* = 11/group). After 4 weeks of nilotinib treatment, livers were removed. Representative images of livers (black arrowheads indicate metastasis) and metastatic index of animals treated daily with vehicle or 50 mg/kg nilotinib (oral administration) (mean ± SEM; *n* = 11/group; *P < 0.05 Student's *t*-test).

Source data are available online for this figure.

CRC. We infected DDR1-depleted HCT116 cells with retroviruses that express wild-type DDR1, kinase-dead (KD) DDR1 or DDR1 T701I at levels similar to those of endogenous DDR1 (Fig 3B) and examined cell invasion in response to nilotinib treatment. Expression of DDR1 or DDR1 T701I, but not of KD DDR1 rescued cell invasion that was lost upon silencing of endogenous DDR1 (Fig 3C, DMSO). This indicates that DDR1 kinase activity is essential for CRC cell invasion. Moreover, cell invasion was inhibited by nilotinib in cells that expressed wild-type DDR1, but not DDR1 T701I (Fig 3C). Similarly, nilotinib prevented liver metastasis formation in nude mice after intrasplenic inoculation of HCT116 cells that expressed wild-type DDR1. Conversely, we observed large metastatic nodules in the liver of mice inoculated with HCT116 cells that expressed DDR1 T701I (Fig 3D). To corroborate DDR1 role in nilotinib mechanism of action, we analysed cell invasion in SW620 cells that

overexpressed or not (mock infection) DDR1. DDR1 promotion of SW620 cell invasion was fully abolished by nilotinib treatment (Fig 3E). Finally, we found that nilotinib treatment could block the growth of already developed metastatic nodules. In this set of experiments, we inoculated animals with SW620-DDR1 cells that express luciferase and started treatment with nilotinib at day 7, when luciferase-positive metastases were already detectable. In this setting, nilotinib prevented liver metastatic progression and decreased ctDNA level of the treated animals (Fig 3F). This effect was confirmed by monitoring the luciferase signal as a surrogate marker of liver tumour burden (Appendix Fig S1). These results highlight an additional important role of DDR1 kinase activity in metastatic growth that can be inhibited by nilotinib.

**Quantitative phosphoproteomic analyses identify BCR as a novel DDR1 substrate**

To gain insight into the signalling pathways targeted by nilotinib, we characterized DDR1 kinase signalling in CRC cells. Western blot analysis of HCT116 cells stimulated with collagen I revealed a slow and moderate, but persistent increase in protein tyrosine phosphorylation content (4–18 h) that correlated with DDR1 TK activation, as measured by DDR1-Tyr792 autophosphorylation. The absence of such molecular response in DDR1-depleted cells (shDDR1) suggests that it might not involve additional collagen receptors (Fig EV2A). On the other hand, DDR1 activation had no effect on MAPK and AKT activities (Fig EV2A), indicating that DDR1 signalling in collagen-stimulated HCT116 cells is RAS-independent. This result was confirmed by the inability of overexpressed DDR1 to stimulate MAPK and AKT activity in SW620 cells (Fig EV2B). We also addressed whether, like observed in lung cancer (Ambrogio *et al*, 2016), DDR1 signalling is regulated by a RAS activity. We found that MAPKs inhibition with a MEK inhibitor had no effect on DDR1 protein expression and collagen-induced DDR1 kinase activation in these CRC cells (Fig EV2C). These data are thus consistent with the RAS-independent nature of DDR1 signalling in CRC.

We then performed a global tyrosine phosphoproteomic analysis by phosphotyrosine peptide immunopurification followed by label-free mass spectrometry-based quantification (Rush *et al*, 2005; Sirvent *et al*, 2015). We identified DDR1 kinase-dependent signalling molecules by comparing the phosphotyrosine profiles of unstimulated HCT116 cells, cells stimulated with collagen I for 18 h, a period that corresponds to the maximal DDR1 kinase activation, and cells stimulated with collagen I and then incubated with nilotinib for 1 h before lysis to induce acute DDR1 inhibition (Fig 4A and B). With this approach, we identified 229 and 289 unique tyrosine-phosphorylated peptides in two independent biological experiments and found 105 common tyrosine-phosphorylated peptides from 83 unique proteins (Dataset EV1). DDR1, BCR and PEAK1 were the only proteins that were reproducibly and significantly phosphorylated by a DDR1 kinase-dependent mechanism (Fig 4C). DDR1 was phosphorylated on Tyr484 of the juxtamembrane domain and on Tyr792 and Tyr796 of the activation loop. This confirms that DDR1 is the major TK activated by collagen I stimulation. The pseudokinase PEAK1, previously reported to be involved in CRC (Wang *et al*, 2010), was phosphorylated on Tyr462 by DDR1, and BCR on Tyr177. BCR

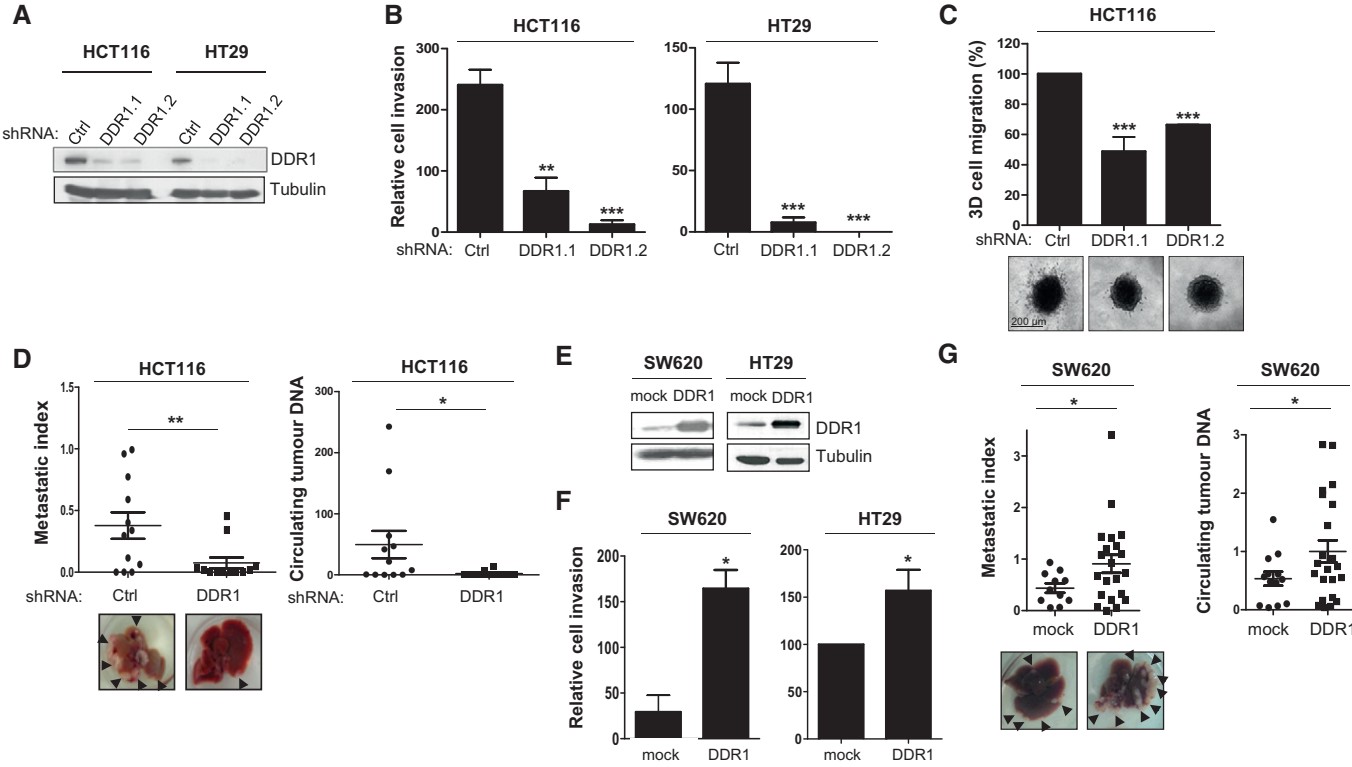

**Figure 2. DDR1 promotes CRC metastasis formation.**

A–D    DDR1 depletion by shRNA inhibits CRC cell invasion and metastasis. (A) DDR1 expression in CRC cells infected with vectors expressing the indicated shRNA was assessed by Western blotting. Invasion of infected CRC cells in Boyden chamber (B) and in collagen I matrix (C) (mean ± SEM; $n = 3$ (B) and $n = 7$ (C); **$P < 0.01$, ***$P < 0.001$ Student's $t$-test). A representative image of cell migration for each cell line is shown. (D) HCT116 cells infected with the indicated vectors were injected in the spleen of nude mice ($n = 12$/group). After 4 weeks, livers were removed. A representative liver image for each group, the metastatic index and the ctDNA level (ng/ml of plasma) for each animal are shown (mean ± SEM; *$P < 0.05$; **$P < 0.01$ Student's $t$-test).

E–G    DDR1 overexpression promotes CRC cell invasion and metastasis formation. (E) DDR1 expression in CRC cells infected with the indicated viruses. (F) Invasion in Boyden chambers of CRC cells that were infected with the indicated viruses (mean ± SEM; $n = 3$; *$P < 0.05$ Student's $t$-test). (G) SW620 cells infected with the indicated viruses were injected in the spleen of nude mice ($n = 12$ for the mock group and $n = 20$ for the DDR1 group). After 4 weeks, livers were removed. A representative liver image for each group, the metastatic index and the relative ctDNA level of each animal are shown (mean ± SEM; *$P < 0.05$ Student's $t$-test).

Source data are available online for this figure.

function has been essentially described in the context of BCR-ABL-transformed leukaemia (Deininger *et al*, 2000). The low number of phosphorylation events was consistent with the moderate increase in tyrosine phosphorylation detected by Western blotting upon collagen I stimulation (Fig EV2A). Surprisingly, we did not detect any of the reported DDR1 effectors in other cell systems, such as SHP2, SHC, CDC42, JAK2 or STAT3 (Juin *et al*, 2014; Leitinger, 2014; Gao *et al*, 2016). Moreover, we identified the additional nilotinib target ABL and its downstream substrates CRK and CRKL by phosphoproteomic analysis; however, their phosphorylation levels were not significantly modified by 18 h of collagen I stimulation, indicating that they are not components of the DDR1 kinase signalling cascade.

Then, to profile DDR1 kinase signalling cascade *in vivo*, we harvested SW620-DDR1 liver nodules from the nude mice treated or not with nilotinib (d7-28) and described in Fig 3F. Western blot analysis revealed that nilotinib treatment did not affect MAPK and AKT activities, consistent with the RAS-independent nature of DDR1 kinase signalling in these metastatic tumours (Fig EV2D). The phosphoproteomic analysis to determine the phosphotyrosine profiles of metastatic nodules from animals treated or not with nilotinib (Fig 4D) allowed us to quantify 430, 482 and 482 unique tyrosine-phosphorylated peptides from three biological replicates and to identify 18 tyrosine-phosphorylated peptides from 14 unique proteins common to all experiments (Dataset EV2). The levels of 12 tyrosine-phosphorylated peptides from 10 unique proteins were decreased upon nilotinib treatment in all three experiments (Fig 4E). We confirmed that DDR1 and BCR were major DDR1 substrates and also identified SHB, PKCδ and FRK (signalling), ITSN2 (trafficking), CRK and Tensin-3 (cytoskeletal-associated proteins), MAGI1 and DLG3 (cell adhesion and polarity proteins, respectively), and the orphan receptor DCBLD2. PEAK1 phosphorylation level on Tyr635 and Tyr531 was decreased in two of the three experiments, suggesting that it may be a DDR1 substrate also *in vivo* (Dataset EV2). We did not detect any of the other known nilotinib targets, suggesting that DDR1 is the main target of this drug in liver metastatic nodules. Overall, these findings indicate that BCR and PEAK1 are important DDR1 signalling substrates in CRC cells.

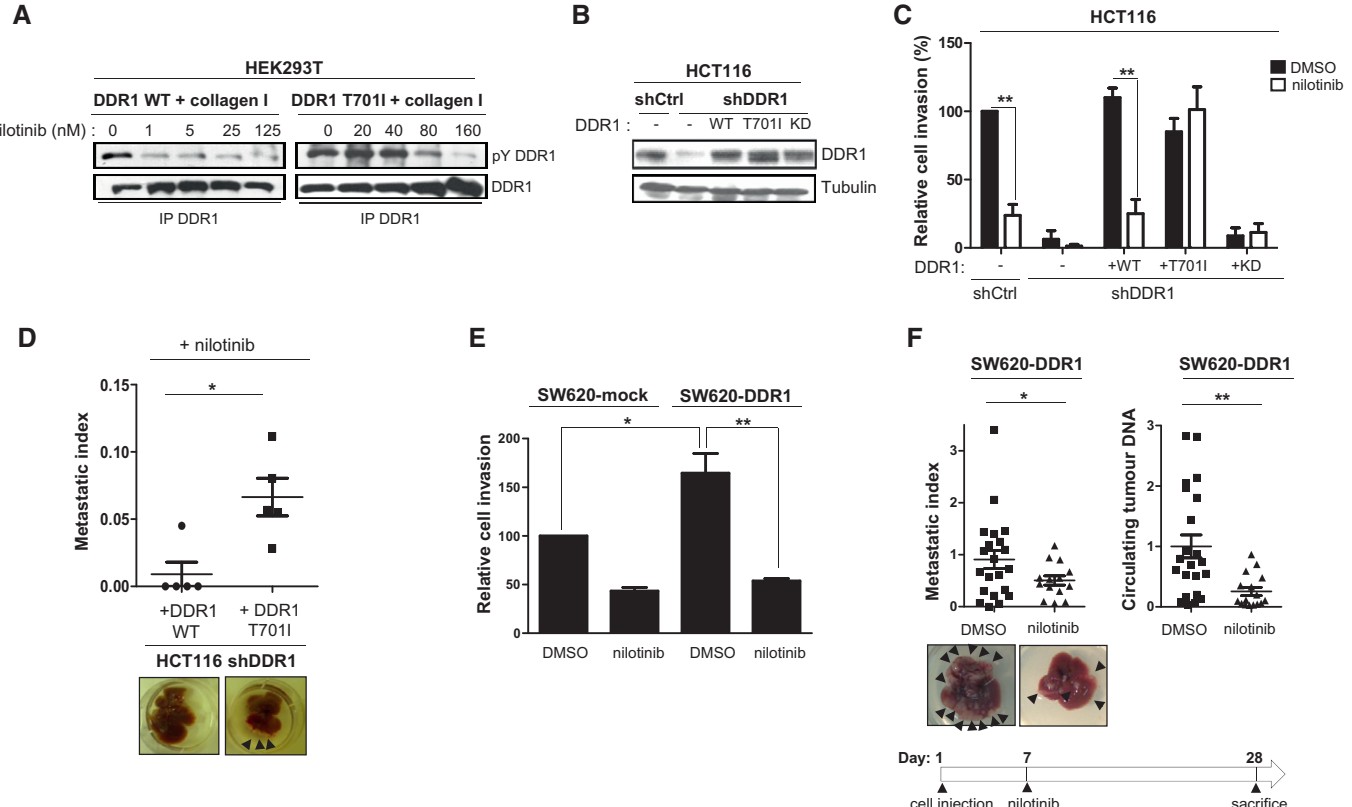

**Figure 3.   Nilotinib anti-tumour activity is mediated by DDR1 kinase inhibition in CRC.**

A    Biochemical analysis of the nilotinib-resistant DDR1 T701I mutant. Tyrosine phosphorylation and DDR1 expression level in HEK293T cells transfected with the indicated DDR1 constructs and stimulated with 40 μg/ml collagen I for 18 h in the presence or not of the indicated concentration of nilotinib.

B–D    Nilotinib anti-tumour activity is abrogated by DDR1 T701I expression in CRC cells. (B) Western blot analysis of DDR1 levels in HCT116 cells expressing the indicated shRNAs and infected with the indicated DDR1 constructs (wild type, WT; T701I; kinase dead, KD). (C) Invasion assays in Boyden chambers of the indicated cell lines after incubation with DMSO or 50 nM nilotinib (mean ± SEM; *n* = 4; **P < 0.01 Student's *t*-test). (D) DDR1-depleted HCT116 cells infected with the indicated DDR1 constructs were injected in the spleen of nude mice (*n* = 5/group) and treated daily with 50 mg/kg nilotinib, starting at day 1 post-injection (oral administration). After 4 weeks, livers were removed. A representative image of liver for each group and the metastatic index of each animal are shown (mean ± SEM; *P < 0.05 Student's *t*-test).

E, F    Nilotinib inhibits DDR1-mediated invasive and metastatic activity of CRC cells. (E) Invasion assays in Boyden chambers of the indicated SW620 cell lines that were incubated with DMSO or 100 nM nilotinib (mean ± SEM; *n* = 3; *P < 0.05; **P < 0.01 Student's *t*-test). (F) After inoculation of SW620 cells that overexpress DDR1 in the spleen, nude mice (*n* = 21 for the DMSO group and *n* = 14 for the nilotinib group) were treated daily with DMSO or 50 mg/kg/d nilotinib as indicated, starting at day 7 post-injection (oral administration). After 4 weeks, livers were removed. A representative image of liver for each group, the metastatic index and the relative ctDNA level of each animal are shown (mean ± SEM; *P < 0.05; **P < 0.01 Student's *t*-test).

Source data are available online for this figure.

## BCR phosphorylation on Tyr177 mediates DDR1 invasive signalling

The finding that BCR is a DDR1 substrate in CRC was unexpected. Therefore, we decided to further characterize its role in DDR1 signalling. We found that in HCT116 cells, collagen I induced slow but persistent phosphorylation of BCR on Tyr177, concomitantly with increased DDR1 kinase activity (Fig EV2A). Conversely, collagen I-mediated BCR phosphorylation was strongly reduced in cells where DDR1 was silenced, or that expressed KD DDR1 or upon incubation with nilotinib (Figs 5A and B, and EV3). This indicates that in CRC cell lines, collagen I-mediated BCR phosphorylation is DDR1 kinase-dependent. In agreement, DDR1 overexpression in SW620 cells induced BCR Tyr177 phosphorylation that was further increased by collagen I stimulation (Fig 5C). Moreover, nilotinib treatment also tended to reduce BCR phosphorylation at Tyr177 in

SW620-DDR1 liver metastases (Fig EV2D). Conversely, nilotinib hardly affected BCR Tyr177 phosphorylation in HCT116 shDDR1 cells that express DDR1 T701I (Fig 5D), suggesting that in CRC cells, this phosphorylation is directly mediated by the DDR1 kinase activity and does not involve ABL. We then found that shRNA-mediated BCR depletion reduced HCT116 cell invasion in Boyden chambers and in collagen I matrix (Fig 5E and F). BCR also regulated DDR1-dependent cell invasion of SW620 cells *in vitro* (Fig 5H). To precisely determine the role of BCR Tyr 177 phosphorylation in CRC cell invasion, we silenced BCR in HCT116 and SW620-DDR1 cells and then infected them with retroviruses expressing wild-type or Y177F BCR (Fig 5G). Expression of wild-type, but not of Y177F BCR restored the invasive capacities of these CRC cells (Fig 5H). Collectively, these results indicate that BCR phosphorylation on Tyr177 plays an essential role downstream of DDR1 to promote CRC cell invasion.

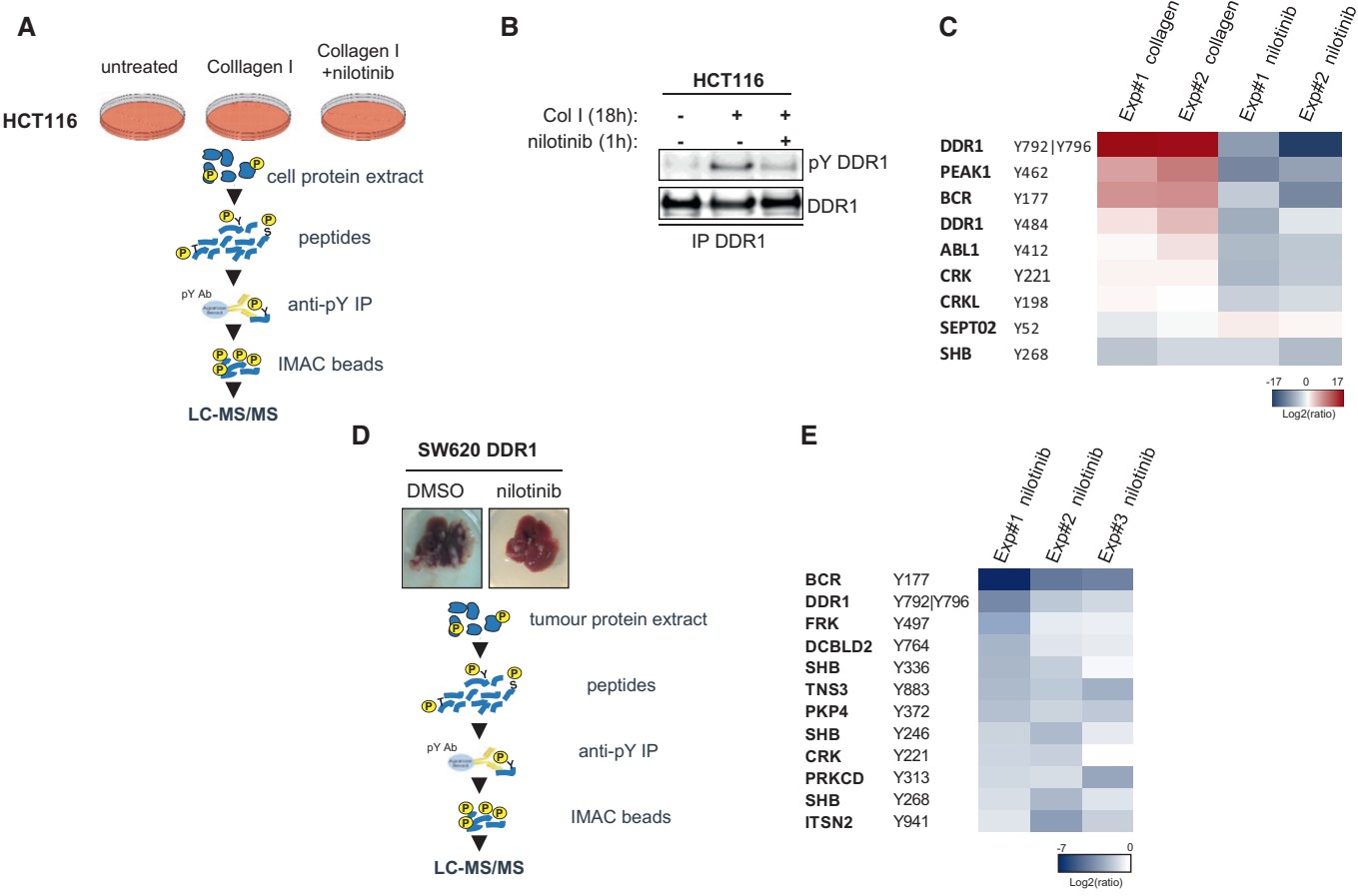

**Figure 4. Quantitative phosphoproteomic analyses identifies BCR as a novel DDR1 substrate.**

A   Workflow of the phosphoproteomic analysis of the DDR1 kinase signalling cascade in HCT116 CRC cells.

B   Western blot analysis of DDR1 tyrosine phosphorylation (pY) in HCT116 cells stimulated with 40 μg/ml collagen I for 18 h and incubated or not with 100 nM nilotinib for 1 h before cell lysis and immunoprecipitation (IP) with an anti-DDR1 antibody.

C   Heatmap of the phosphotyrosine peptides in HCT116 cellular extracts for which the level was consistently modulated upon collagen I stimulation and by nilotinib treatment (two biological replicates).

D   Workflow of the phosphoproteomic analysis of the DDR1 kinase signalling cascade in liver metastases of the nude mice inoculated with DDR1 overexpressing SW620 cells described in Fig 3F.

E   Heatmap of phosphotyrosine peptides in liver metastasis protein extracts (nude mice described in Fig 3F) that were consistently reduced upon nilotinib treatment (three biological replicates).

Source data are available online for this figure.

## DDR1 phosphorylates BCR to support β-catenin transcriptional activity

BCR phosphorylation at Tyr177 has been linked to RAS activation in CML cells; however, this mechanism may not operate in collagen I-stimulated CRC cells because DDR1 does not regulate the RAS pathway. Interestingly, Ress and Moelling reported a BCR negative role on Wnt signalling in CRC cells, which can be alleviated by its phosphorylation on Tyr177 (Ress & Moelling, 2005). Mechanistically, BCR interacts with β-catenin in the cytoplasm to prevent its nuclear localization and expression of its target genes. BCR Tyr177 phosphorylation disrupts this interaction, resulting in increased β-catenin nuclear activity (Ress & Moelling, 2005). We thus hypothesized that, by inducing BCR Tyr177 phosphorylation, DDR1 suppresses this negative regulatory loop to support the β-catenin

signalling necessary for CRC cell motility. To test this hypothesis, we analysed DDR1 effect on β-catenin/TCF-dependent transcriptional activity in HEK293 and HCT116 cells using the TCF reporter TOPflash and the mutant reporter FOPflash as a control. Expression of DDR1 strongly promoted β-catenin/TCF-mediated transactivation of TOPflash in HEK293 and also in HCT116 cells (Fig 6A and B). In HEK293 cells, this response was inhibited by nilotinib and by KD DDR1 expression (Fig 6A), demonstrating the essential role of DDR1 kinase activity in this pathway. We then examined DDR1 effect on the induction of specific β-catenin target genes involved in CRC cell motility (Jackstadt et al, 2013; Liu et al, 2016). DDR1 expression in SW620 cells increased the transcript levels of MYC, FRA1, and JUN compared with control (mock infection) (Fig 6C). This upregulation required DDR1 TK activity because it was inhibited by nilotinib or expression of KD DDR1 (Fig 6C). Conversely,

DDR1 silencing in HCT116 cells significantly reduced the expression levels of these target genes compared with control (shCtrl) (Fig 6D). DDR1 depletion also reduced the transcript level of canonical Wnt target genes *CD44*, *CCND1*, *LGR5* and *AXIN2* involved in CRC stem cell properties (Vanharanta & Massague, 2013); however, it had no effect on *ASCL2* and *SLC12A2* (Fig EV4A). This suggests a role for endogenous DDR1 in β-catenin oncogenic activity towards a subset of Wnt target genes in CRC cells. To ascertain our model, we also addressed the role of BCR phosphorylation on DDR1-induced β-catenin nuclear activation. Immunofluorescence analysis in SW620 cells shows that DDR1 overexpression increased nuclear localization of active β-catenin (Fig EV4B). This response was further increased by collagen I stimulation and to a level similar to the one obtained with Wnt3A (Figs EV4B and 6E). Importantly, this DDR1 response required its TK activity and phosphorylation of BCR on Tyr177 as nuclear β-catenin level was strongly inhibited in cells treated with nilotinib or expressing the BCR Y177F mutant (Fig 6E). We next addressed the *in vivo* relevance of our model by investigating the effect of DDR1 activity on the oncogenic level of β-catenin in liver metastatic tumours obtained in Fig 3F. Immunohistochemistry staining of liver sections shows that DDR1 expression significantly increases the level of nuclear and active β-catenin in CRC cells, specifically at the front of the tumour, while nilotinib treatment significantly reduces this molecular response (Figs EV4C and 6F).

### A BCR/β-catenin signalling is involved in DDR1 invasive activity

We next explored the functional relevance of our hypothesis on CRC cell invasion. If our model is true, then BCR may sequester β-catenin in the cytoplasm and functions as a negative regulator of CRC cell migration, unless phosphorylated on Tyr177. Consistently, in the absence of collagen, the level of BCR pTyr177 in CRC cells (HCT116, SW620 DDR1) is low, BCR depletion increased cell migration in Boyden chamber assays (Fig EV5A and B). This effect was abolished by BCR and BCR Y177F expression, highlighting the phosphorylation-independent BCR anti-migratory activity (Fig EV5B). In contrast, BCR was phosphorylated on Tyr177 upon collagen stimulation (collagen I or collagen IV present in Matrigel) (Fig EV5A) and its depletion reduced CRC cell invasion in Matrigel (Fig 5H). Invasion was restored by BCR but not BCR Y177F expression (Fig 5H), highlighting the phosphorylation-dependent BCR invasive activity. We next addressed the role of β-catenin on DDR1 invasive activity. Pharmacological inhibition of β-catenin signalling with IWR-1-endo, a Wnt signalling inhibitor that promotes β-catenin proteasomal degradation, significantly reduced DDR1-mediated cell invasion of SW620-DDR1 cells, but not of mock-transfected SW620 cells, compared with vehicle (DMSO) (Fig 6G). Overall, these findings support our model in which DDR1 phosphorylates BCR to support β-catenin oncogenic signalling important for CRC cell invasion.

### DDR1 expression is associated with shorter overall survival in patients with CRC

We next examined the clinical relevance of our findings using CRC specimens. DDR1 somatic mutation or gene amplification is not frequent in CRC (< 3% in cbioportal.org); however, transcriptomic

analysis performed from 143 stage IV CRC patients CRC (Del Rio *et al*, 2017) reveals that a high level of DDR1 transcript is associated with a shorter relapse-free and overall survival (Fig 7A). Cox proportional hazard models showed that DDR1 expression level is an independent marker of poor prognosis in stage IV patients and is not correlated with already described variables like tumour location, grade, number of metastatic sites and CMS transcriptional subtype (Appendix Table S1). In particular, there is no significant difference in the expression level of DDR1 between the four CMS subtypes (Guinney *et al*, 2015; Appendix Fig S2). As DDR1 promoted CRC metastasis formation in our experimental models, we also examined DDR1 activity in CRC biopsies. However, due to the labile nature of DDR1 tyrosine phosphorylation (Fig EV1C), we failed to detect DDR1 activity by immunohistochemical analysis (not shown). Nevertheless, we could detect DDR1 activity in a small group of samples from patients with mCRC, composed of matched healthy tissue, primary tumour and metastatic lesions, which were quickly frozen after resection (Del Rio *et al*, 2007). We deduced the specific DDR1 activity from the relative level of DDR1 tyrosine phosphorylation measured by Western blotting. We observed moderate DDR1 activation in matched primary tumour and healthy tissue samples. On the other hand, DDR1 phosphorylation was strongly increased in the corresponding metastatic lesions (Fig 7B). These findings support the notion that DDR1 activity promotes CRC metastasis formation and that its expression level correlates with the tumour aggressiveness.

### DDR1 signalling inhibition reduces invasion and metastasis of patient-derived CRC cell lines

Finally, we examined DDR1 invasive activity and its inhibition by nilotinib in patient-derived CRC cell lines we recently generated from metastatic tumours and circulating CRC cells in blood samples (Grillet *et al*, 2017). These include the CPP19 and CPP30 cell lines that have been freshly established from liver metastasis biopsies that express mutated KRAS and wild-type KRAS/BRAF, respectively, and the CTC44 and CTC45 cell lines derived from circulating tumour cells (CTC) that express mutated BRAF from two chemotherapy-naive patients with metastatic CRC (stage IV). We could not derive CTC lines from patients with lower stage CRC or who had been treated by chemotherapy. Consistent with the essential role of CTCs in metastasis development in distant organs, our CTC lines show self-renewal capacities and metastatic properties when injected in the spleen of nude mice (Grillet *et al*, 2017). We found that DDR1 was well expressed and active in these four cell lines, as indicated by the increase in DDR1 autophosphorylation and BCR Tyr177 phosphorylation upon collagen I stimulation (Fig 7C and F). DDR1 silencing decreased their invasive potential in Boyden chambers and nilotinib had a significant inhibitory effect on DDR1 signalling and cell invasion, except in CTC45 cells (Fig 7D, E, G and H), consistent with a DDR1 kinase-dependent role in CRC cells. Finally, we analysed the anti-metastatic activity of nilotinib on these patient-derived CRC cells. In this set of experiments, we inoculated animals with wild-type KRAS/BRAF CPP30 cells and started nilotinib treatment at day 6, when CRC cells are expected to develop micrometastases. In this setting, nilotinib treatment reduced liver metastatic progression (Fig 7I). These results highlight the important role of DDR1 kinase activity in CRC metastatic growth that can be inhibited by nilotinib, independently of the KRAS/BRAF genotype.

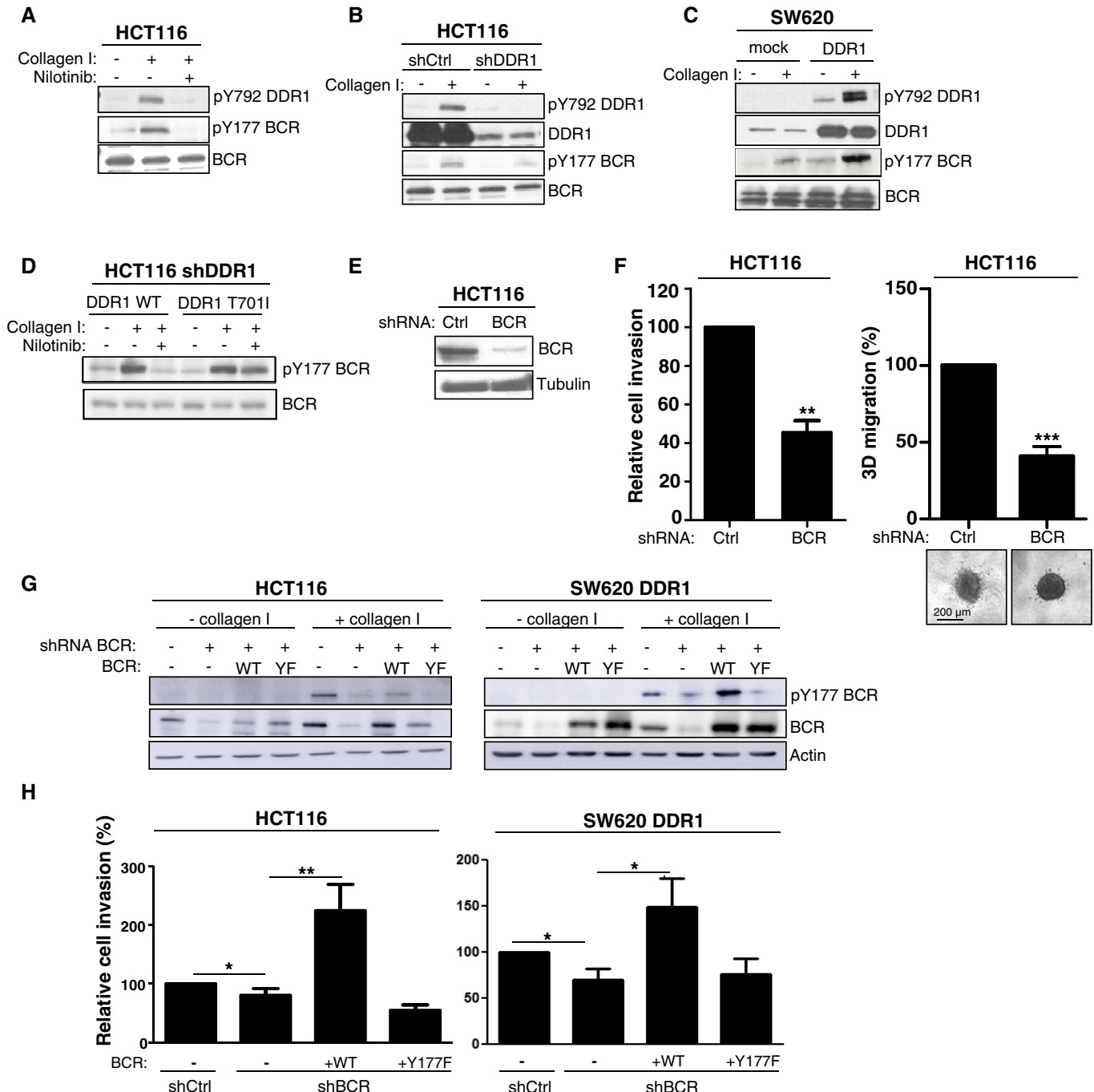

**Figure 5. BCR phosphorylation on Tyr177 mediates DDR1 invasive signalling.**

A–D BCR is a novel DDR1 substrate in CRC cells. (A) Incubation with 100 nM nilotinib and (B) shRNA-mediated silencing of DDR1 reduce BCR phosphorylation at pTyr177 (pTyr177-BCR) induced by collagen I stimulation (40 μg/ml for 18 h). (C) DDR1 overexpression in SW620 cells increases pTyr177-BCR level induced by collagen I stimulation (40 μg/ml for 18 h). (D) pTyr177-BCR induced by collagen I stimulation does not require ABL-like activities. pTyr177-BCR levels were assessed in HCT116 cells in which DDR1 was silenced by shRNAs and transfected with the indicated DDR1 constructs, stimulated or not with collagen I (40 μg/ml for 18 h) and incubated or not with 100 nM nilotinib.

E, F BCR modulates HCT116 cell invasion. (E) The level of BCR depletion upon infection with viruses expressing the indicated shRNAs was checked by Western blotting. (F) Invasion assays in Boyden chamber (left) and in collagen I matrix (right) of the indicated HCT116 cell lines (mean ± SEM; $n = 3$ and $n = 7$, respectively; **$P < 0.01$; ***$P \leq 0.001$ Student's $t$-test). A representative image of cell migration for each condition is shown.

G, H Tyr177-BCR regulates DDR1 invasive activity. (G) BCR and pTyr177-BCR levels were assessed in CRC cells expressing the indicated shRNAs and infected with viruses that express the indicated BCR constructs (wild type, WT; Y177F, YF) and stimulated or not with collagen I (40 μg/ml for 18 h). (H) Invasion assays in Boyden chambers of the indicated cell lines (mean ± SEM; $n = 4$; *$P < 0.05$; **$P < 0.01$ Student's $t$-test).

Source data are available online for this figure.

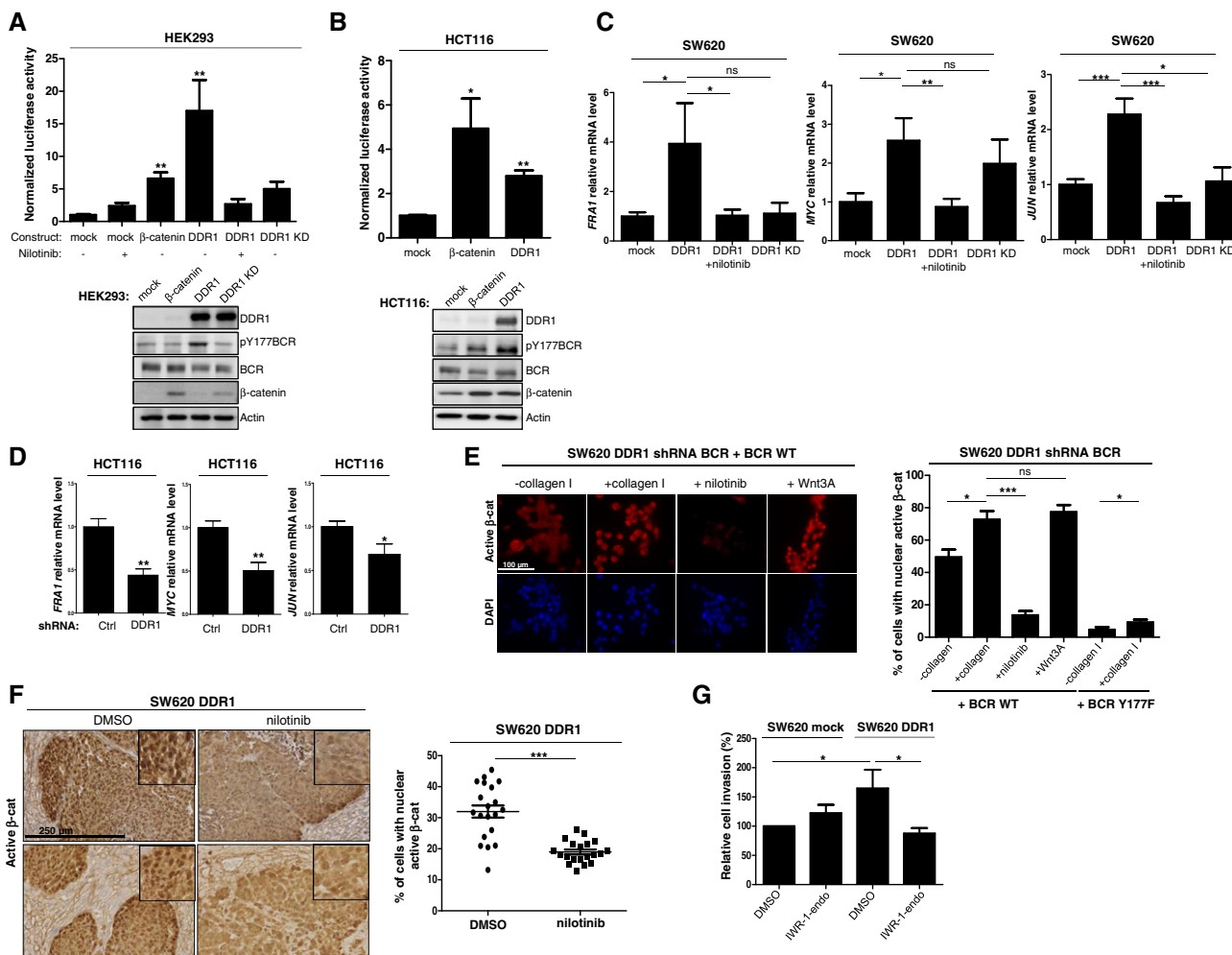

**Figure 6.  DDR1 kinase signalling increases β-catenin transcriptional activity.**

A, B   DDR1 kinase activity increases β-catenin transcriptional activity. (A) HEK293 cells and (B) HCT116 cells were transfected with TOPflash, mutant control FOPflash,
       Renilla luciferase and the indicated constructs and incubated (+) or not (−) with 100 nM nilotinib. The luciferase activity normalized to control condition is shown
       (mean ± SEM; *n* = 2 independent experiments with three replicates; \*P < 0.05; \*\*P < 0.01 Mann–Whitney test). Western blot analyses confirmed the expression of
       the different DDR1 variants and of β-catenin.
C      DDR1 kinase activity increases the transcript level of selected β-catenin target genes in CRC cells. Relative transcript level of the indicated genes in SW620 cells
       infected with viruses that express the indicated DDR1 variants and incubated or not with 100 nM nilotinib as shown (mean ± SEM; *n* = 2 independent
       experiments with three replicates; ns: not significant; \*p < 0.05 \*\*P < 0.01; \*\*\*P < 0.001 Mann–Whitney test).
D      DDR1 regulates the transcript level of selected β-catenin target genes in CRC cells. Relative transcript level of the indicated genes in HCT116 cells infected with viruses
       that express that indicated shRNAs (mean ± SEM; *n* = 2 independent experiments with three replicates; ns: not significant; \*P < 0.05; \*\*P < 0.01; Mann–Whitney test).
E      DDR1 signalling increases β-catenin nuclear activity in CRC cells. Immunofluorescence analysis of active β-catenin in SW620 cells infected with indicated viruses
       and stimulated overnight or not with collagen I (50 μg/ml), Wnt3A (200 ng/ml) and treated or not with nilotinib (100 nM) as indicated. A representative example
       (left panel) and quantification (right) of cells with active nuclear β-catenin (mean ± SEM; *n* = 3; ns: not significant; \*P < 0.05 \*\*\*P < 0.01; Mann–Whitney test).
       Scale bar: 100 μm.
F      DDR1 signalling increases β-catenin nuclear activity in metastatic CRC. IHC analysis of active β-catenin level in experimental metastatic tumours described in
       Fig 3F. A representative example (left) and quantification (right) of CRC cells with active nuclear β-catenin (mean ± SEM; *n* = 5 tumours per group with 4 fields
       per tumour; ns: not significant; \*P < 0.05 \*\*\*P < 0.001; Mann–Whitney test). Scale bar: 250 μm.
G      DDR1 invasive activity in CRC cells requires proper β-catenin expression. Invasion assays in Boyden chambers of SW620 cells infected with the indicated viruses
       and incubated with DMSO or with 0.1 mM of the β-catenin pharmacological inhibitor IWR-1-endo (mean ± SEM; *n* = 3; \*P < 0.05 Student's *t*-test).

Source data are available online for this figure.

## Discussion

Here, we identified DDR1 role in CRC metastasis formation. DDR1 promotes invasion and metastatic behaviour of CRC cells in nude mice and its overexpression potentiates these properties. DDR1 also

regulates invasiveness of patient-derived CRC cell lines from metastatic tumours and circulating CRC cells and its expression level is associated with shorter overall survival in patients with mCRC. These results corroborate previous reports showing DDR1 pro-invasive role in various cell lines derived from tumours of epithelial

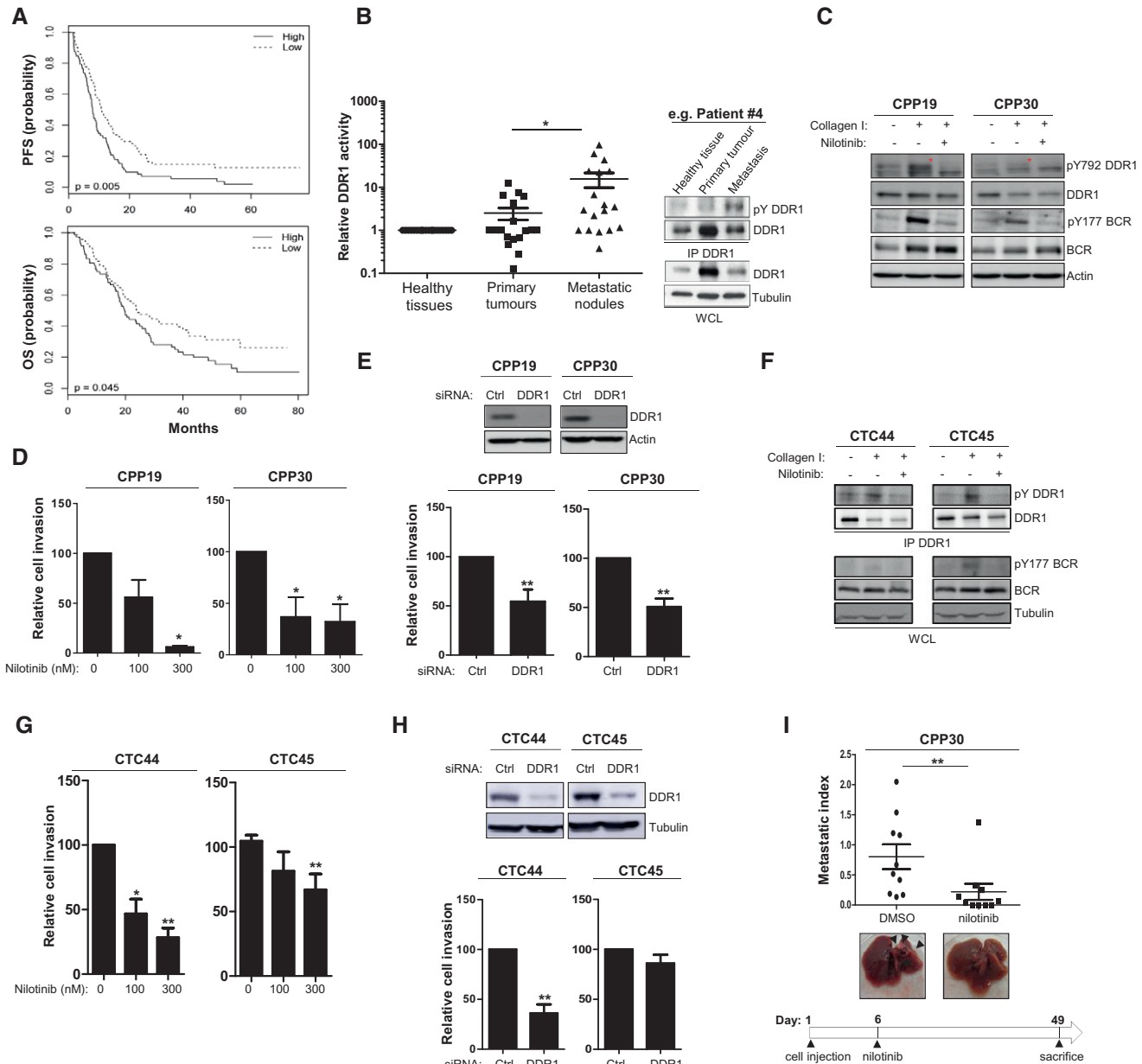

**Figure 7. Nilotinib inhibits DDR1 invasive activity of patient-derived CRC cells.**

A    Patients with CRC showing high DDR1 expression have shorter progression-free survival (PFS) and overall survival (OS). Kaplan–Meier analysis using data from 143 patients with stage IV CRC subdivided according to the tumour DDR1 expression level (high/low).

B    Increased DDR1 activity in metastatic nodules from patients with CRC. DDR1 activity was evaluated based on the relative DDR1 tyrosine phosphorylation level measured by immunoprecipitation (IP) followed by Western blotting of protein lysates of matched healthy tissue, primary tumour and metastatic lesions from 18 patients with CRC. A representative example for one patient (upper panels) and the relative level of DDR1 tyrosine phosphorylation in each patient quantified by ImageJ (lower histogram) are shown (mean ± SEM; *P < 0.05 Student's t-test).

C    DDR1 activity in patient-derived CRC lines stimulated or not with collagen I (40 μg/ml for 18 h) and incubated or not with 100 nM nilotinib (red asterisks indicate DDR1 activation).

D, E    DDR1 invasive activity in patient-derived CRC lines. Cell invasion was assessed in Boyden chambers after incubation with the indicated doses of nilotinib (D) or transfection with the indicated siRNAs (mean ± SEM; left panels n = 3 and right panels n = 6; *P < 0.05; **P < 0.01 Student's t-test). (E) Level of DDR1 depletion upon siRNA transfection.

F    DDR1 signalling in patient-derived CRC cell lines stimulated or not with collagen I (40 μg/ml for 18 h) and treated or not with nilotinib (100 nM) as shown.

G, H    DDR1 invasive activity in patient-derived CRC cell lines. Cell invasion was assessed in Boyden chambers after incubation with the indicated doses of nilotinib (G) or transfection with the indicated siRNAs (mean ± SEM; n = 4; *P < 0.05; **P < 0.01 Student's t-test). (H) Level of DDR1 depletion upon siRNA transfection.

I    Nilotinib inhibits metastatic activity of CPP30 cells. After inoculation of CPP cells in the spleen, nude mice (n = 10/group) were treated daily with DMSO or 50 mg/kg nilotinib (i.p.) as indicated, starting at day 6 post-injection. After 49 days, livers were removed. A representative image of liver for each group and the metastatic index of each animal are shown (mean ± SEM; *P < 0.05 Student's t-test).

Source data are available online for this figure.

origin and DDR1 metastatic function in lung and breast cancer (Valencia *et al*, 2012; Gao *et al*, 2016), and highlights a conserved function of DDR1 in invasive tumours. Importantly, our results establish the central role of DDR1 TK activity in this malignant process, as indicated by the loss of such function in cells that express DDR1 harbouring a kinase-dead mutation or upon specific pharmacological inhibition with nilotinib. The clinical relevance of this DDR1 activity is supported by the strong increase in DDR1 phosphorylation in metastatic CRC lesions compared with matched primary tumours and healthy tissues. As DDR1 also activates kinase-independent signalling cascades to induce collective migration or extracellular matrix degradation of tumour cells and metastasis reactivation of disseminated tumour cells (Hidalgo-Carcedo *et al*, 2011; Juin *et al*, 2014; Gao *et al*, 2016), we propose that DDR1 induces both kinase-dependent and -independent pathways to fully promote metastasis formation. However, some DDR1 kinase-independent mCRC may be expected, since nilotinib does not impact on the invasive abilities of all CRC cells tested (for instance DLD1 and SW48).

We also identified a potentially important mechanism by which DDR1 TK activity promotes cancer cell invasion and metastasis formation (Fig 8). Our phosphoproteomic analyses show that, by phosphorylating BCR on Tyr177, DDR1 alleviates a negative regulatory loop on β-catenin signalling to sustain its oncogenic activity (Ress & Moelling, 2005). In agreement, DDR1 activity induces expression of β-catenin target genes that are important for cell motility, such as *JUN*, *FRA1*, *CD44* and *MYC*, and DDR1 invasive activity requires intact β-catenin activity in CRC cells. Due to the major role of the Wnt/β-catenin pathway in CRC, we propose that DDR1 acts by supporting β-catenin oncogenic activity upon adhesion to collagen matrix to sustain cell migration, survival or renewal. We also found that BCR is a major DDR1 substrate. This unexpected result raises the possibility that DDR1 phosphorylates BCR also in CML and that BCR role in human cancer may not be restricted to leukaemia. Our phosphoproteomic analyses suggest that DDR1 may activate additional oncogenic pathways. For instance, we identified PEAK1 as an additional DDR1 substrate that is phosphorylated on novel sites in CRC cells. This pseudokinase and scaffolding protein is a novel growth factor receptor signalling component (Zheng *et al*, 2013) and a key regulator of cell spreading and migration (Croucher *et al*, 2013). In CRC, it promotes anchorage-independent growth and tumour development in xenografted nude mice (Wang *et al*, 2010). We found that PEAK1 also regulates the invasive properties of CRC cells (not shown). Our results also suggest an additional role of BCR in this process, since its depletion affects DDR1 invasive activity without impacting on beta-catenin nuclear activity. Finally, a previous study linked DDR1 to Notch signalling in lung cancer (Ambrogio *et al*, 2016) and Notch also acts as an oncogene in metastatic CRC (Sonoshita *et al*, 2011). Through induction of DAB1 expression, Notch induces a DAB1/ABL/Rho GEF TRIO1 signalling cascade to promote invasive colorectal tumours (Sonoshita *et al*, 2015). However, we did not found any of these signalling components in our phosphoproteomic analyses, suggesting that DDR1 does not intersect this signalling cascade. Nevertheless, we cannot exclude that DDR1 interacts with the Notch pathway through an alternative, not yet described mechanism.

Our results also suggest that DDR1 acts on several aspects of CRC liver metastasis formation. First, DDR1 could promote local invasion of primary tumour cells and the invasive properties of

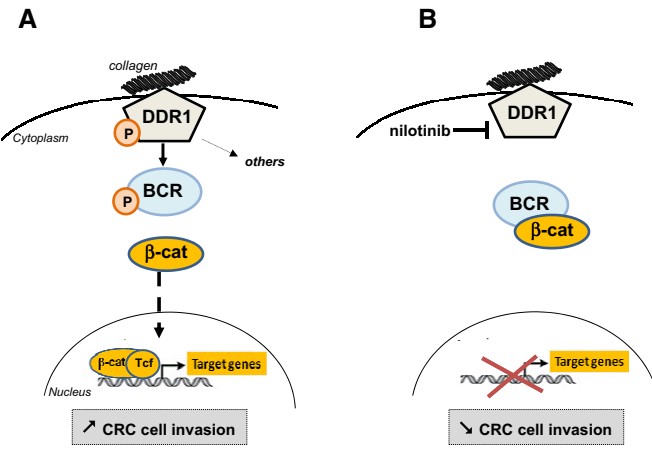

**Figure 8.  Proposed model for kinase-dependent DDR1 invasive activity in CRC cells.**

A  Collagen-stimulated DDR1 activation induces the phosphorylation of BCR on Tyr177, which disrupts BCR/β-catenin interaction. This signalling cascade results in an increased β-catenin nuclear activity leading to expression of target genes necessary for cell invasion.

B  Inhibition of DDR1 kinase activity with nilotinib decreases CRC cell invasion by reducing this β-catenin-dependent signalling cascade necessary for cell invasion.

disseminated CRC cells, which is essential for metastasis formation. We hypothesize that DDR1 activates, via a kinase-independent mechanism, Tuba and CDC42 to induce early proteolysis-based invasion (Juin *et al*, 2014), while its kinase activity is required to sustain, through a β-catenin-dependent mechanism, cell motility in a collagen-rich environment. Recently, it has been shown that DDR1 is involved in myosin-dependent collagen contraction and this function might also contribute to efficient CRC cell invasion in a collagen-rich matrix (Coelho *et al*, 2017). DDR1 activity could also promote homing of disseminated tumour cells in the liver upon collagen deposition to support β-catenin-dependent cell survival, as documented for other extracellular matrix components, such as tenascin C in metastatic breast cancer (Oskarsson *et al*, 2011). This idea is experimentally supported by the strong reduction in liver metastasis development observed following early DDR1 inhibition by nilotinib treatment when most inoculated CRC cells have already reached the liver and the strong diminution of nuclear β-catenin activity at the front of the metastatic tumour. Finally, DDR1 activity may play a role in metastasis growth, as suggested by the anti-tumour effect and reduction in ctDNA levels and luciferase signals in animals with detectable metastases treated with nilotinib. While DDR1 has been involved in the growth of specific CRC cell line (Hu *et al*, 2014), we found that its expression level and its inhibition by nilotinib does not significantly modulate the *in vitro* growth of the tested CRC cell lines in standard conditions (Appendix Fig S3). It is possible that DDR1 growth effect is context-dependent. In support of this hypothesis, in metastatic tumours, DDR1 could contribute to increase the β-catenin activity necessary for CRC metastasis growth. Altogether, these observations indicate that DDR1 is an important therapeutic target in mCRC.

Finally, our report reveals that nilotinib could be used to block DDR1 TK activity in CRC. Nilotinib is a high-affinity TKI for DDR1

and ABL and is currently used in the clinic to treat patients with CML. Our results suggest that pharmacological inhibition of DDR1 by nilotinib could define an effective strategy to inhibit canonical DDR1 signalling because its kinase activity is tightly regulated by phosphatases in CRC. Moreover, additional nilotinib targets have been involved in CRC, although they were not recovered in our phosphoproteomic analyses. For instance, the EphB-ABL signalling pathway is associated with intestinal tumour initiation and growth (Kundu et al, 2015), and ABL is a component of the Notch invasive signalling cascade in CRC (Sonoshita et al, 2015). PDGF receptor blockade also affects the metastatic behaviour of specific CRC cell lines through modulation of the tumour microenvironment (Yuge et al, 2015). Therefore, inhibition of these TKs could contribute to the observed anti-metastatic activity of nilotinib in CRC. Our results also suggest that, by dampening DDR1-dependent β-catenin activity in CRC, nilotinib reduces tumour cell dissemination and prevents their capacity to target distant organs in patients with CRC. Nilotinib could also reduce CRC metastatic growth, thus contributing to the reduction in the metastatic burden. As DDR1 overexpression increases the metastatic behaviour of CRC cells and high DDR1 expression in CRC specimens is associated with shorter overall survival, DDR1 expression or activity level could predict nilotinib response in patients with advanced CRC.

In conclusion, we identified DDR1 as a RAS-independent therapeutic target in CRC. Our results suggest that DDR1 inhibition with nilotinib could be used in patients with mCRC, including those who are refractory to anti-EGFR antibodies. They also suggest that aberrant DDR1 signalling may contribute to the high level of Wnt oncogenic activity observed in some mCRC and predict that combinations of nilotinib with agents targeting the Wnt/β-catenin pathway may be effective against these tumours. It will be now important to determine whether the combination of DDR1 inhibition with these therapeutic strategies could be effective in patients with metastatic CRC.

# Materials and Methods

### Antibodies and reagents

The following antibodies (sources indicated between brackets) were used: anti-DDR1 C-20, anti-β-catenin E-5 (Santa Cruz Biotechnology); anti-ERK1/2, anti-ERK1/2 pT202/Y204, anti-AKT, anti-AKT pS473 anti-pY792 DDR1, anti-non-phospho (active) β-catenin (Ser33/37/Thr41), anti-BCR N-20 and anti-pTyr clone pY1000 Sepharose bead conjugated (PTM Scan, Cell Signaling Technology); anti-β-actin (Sigma-Aldrich); anti-active-β-catenin 8E7 (Merck Millipore); anti-pY177 BCR (Abgent); anti-tubulin (gift from N. Morin, CRBM, Montpellier, France); anti-pTyr 4G10 (gift from P. Mangeat, CRBM, Montpellier, France); anti-rabbit IgG-HRP and anti-mouse IgG-HRP (GE Healthcare); goat anti-mouse Alexa Fluor 594 (Thermo Fisher Scientific). Nilotinib was a generous gift from Novartis, and collagen I rat tail was from Invitrogen and IWR-1-endo from SelleckChem.

### Plasmids

The pIRESneo construct to express the human DDR1 isoform 1a was from C. Laschinger (Toronto) and was subcloned in the pMX-pS-CESAR retroviral vector. The DDR1 T701I and DDR1 K655A mutants were obtained by PCR using the QuickChange Site-Directed Mutagenesis Kit (Stratagene) and the following oligonucleotides: 5′-gaccccctctgcatgattattgactacatggaga-3′ and 5′-cctttgctggtagctgtcgc gatcttacggccaga-3′. Human BCR in the pKH3 vector from Addgene was cloned in the pBABEhygro vector (Addgene). The BCR Y177F mutant was obtained by PCR using the QuickChange Site-Directed Mutagenesis Kit (Stratagene) and the following oligonucleotide: 5′-cgccgagaagcccttcttcgtgaacgtc-3′. The shRNA sequences used are listed below and were cloned in the pSiren-retroQ or pRETRO-SUPER.neo.GFP retroviral vectors according to the manufacturer's instructions (Clontech). Constructs to express a scrambled shRNA (shmock) or a shRNA directed against luciferase (shCtrl) were used as negative controls. Silent mutations in the DDR1 and BCR constructs were introduced to render them insensitive to DDR1 and BCR shRNAs using the QuickChange Site-Directed Mutagenesis Kit (Stratagene) and the following oligonucleotides: 5′-atggctgcctctgga gagacggcctgctctcctacaccgcccctgtgg-3′ and 5′-gagcgcggcctggtgaaagtga atgataaagaggtgtcggaccg-3′.

| shRNA | Sequence | Vector |
| --- | --- | --- |
| shCtrl | Clontech | pSiren-retroQ |
| shDDR1.1 | gggatggactcctgtctta | pSiren-retroQ |
| shDDR1.2 | agatggagtttgagtttgac | pSiren-retroQ |
| shMock | gacactcggtagtctatac | pSUPER.retro.neo.GFP |
| shBCR | ggtcaacgacaaagaggtg | pSUPER.retro.neo.GFP |

### Cell cultures, retroviral infections and transfections

Cell lines (ATCC, Rockville, MD) were cultured at 37°C and 5% $CO_2$ in a humidified incubator in Dulbecco's modified Eagle's medium (DMEM) GlutaMAX (Invitrogen) supplemented with 10% foetal calf serum (FCS), 100 U/ml of penicillin and 100 μg/ml of streptomycin (growth medium). Human patient-derived metastatic (CPP) and circulating tumour (CTC) CRC cell lines were obtained from J. Pannequin (IGF, Montpellier) and cultured as previously described (Grillet et al, 2017). Retroviral infections were described in Chevalier et al (2016), and stable cell lines were obtained by selection with 1 μg/ml puromycin (Sigma-Aldrich) or 400 μg/ml hygromycin B Gold (InvivoGen) or by fluorescence-activated cell sorting. Transient plasmid transfections in HEK293T cells were performed with the jetPEI reagent (Polyplus-transfection) according to the manufacturer's instructions. For siRNA transfection, $2.10^5$ cells were seeded in 6-well plates and transfected with 20 nmol of siRNA and 9 μl of Lipofectamine RNAi Max according to the manufacturer's protocol (Thermo Fisher). A scrambled siRNA (siMock, non-targeting siRNA#1 from Dharmacon) was used as a negative control. siRNAs against human DDR1 and human BCR were purchased from Life Technologies (Stealth siRNAs HSS187879 DDR1 and s1949 BCR).

### Standard proliferation assays

50,000 cells were seeded in 24-well plates in DMEM containing 2% FCS and then fixed every day in trichloroacetic acid (TCA) solution. Standard cell growth was measured by sulforhodamine B staining (Sigma-Aldrich) according to the manufacturer's instructions.

## Immunocytochemistry

Cells were grown in Nunc Lab-Tek II chamber slide and fixed 5 min with 4% paraformaldehyde and 10 min with ice-cold methanol. Permeabilization and blocking were done using 0.3% Triton X-100-0.3% BSA in PBS for 2 h at room temperature. Cells were incubated with primary antibodies overnight at 4°C (dilution 1:1,000). Fluorescent secondary antibodies (dilution 1:3,000) were used for 1 h at room temperature, and the slides were mounted with DAPI Fluoromount-G (SouthernBiotech, Birmingham, USA). Images were acquired on a Zeiss Axioimager Z2 microscope, processed and analysed using ImageJ software.

## Immunohistochemistry

Formalin-fixed and paraffin-embedded xenograft tissues were cut into 5-μm sections. After deparaffinization and rehydration, tumour sections were subjected to antigen unmasking treatment for 20 min in Tris-EDTA buffer pH 9.0 (10 mM Tris Base, 1 mM EDTA Solution, 0.05% Tween 20). Slides were then treated with BLOXALL Blocking Solution (Vector Laboratories) for 10 min to quench endogenous peroxidase activity, permeabilized for 10 min in PBS-Triton 0.3% and blocked with 2.5% normal horse serum in PBS for 20 (ImmPRESS reagent kit, Vector Laboratories, Burlingame, CA). Sections were incubated at 4°C overnight with the primary antibodies (dilution 1:500). Secondary antibodies (ImmPRESS reagent kit, Vector Laboratories) were incubated for 30 min at room temperature. Detection was performed with ImmPACT DAB kit (Vector Laboratories). Counterstaining with haematoxylin was done for 4 min. Sections were scanned using Nanozoomer Slide Scanner (Hamamatsu City, Japan), and quantifications were performed using ImageJ software by calculating the percentage of tumour cells with a positive nuclear active β-catenin staining.

## Biochemistry

Immunoprecipitation and immunoblotting were performed as described in Sirvent *et al* (2010). Briefly, cells were lysed at 4°C with lysis buffer (20 mM Hepes pH7.5, 150 mM NaCl, 0.5% Triton X-100, 6 mM β-octylglucoside, 10 μg/ml aprotinin, 20 μM leupeptin, 1 mM NaF, 1 mM DTT and 100 μM sodium orthovanadate). Immunoprecipitation was performed with 500 μg proteins and 2 μg of the specific antibody. Immunoprecipitates or 20–50 μg of whole cell lysates was loaded on SDS–PAGE gels and transferred onto Immobilon membranes (Millipore). All primary antibodies were used at 1:1,000 dilution and secondary HRP-conjugated antibodies at 1:4,000 dilution. Detection was performed using the ECL System (Amersham Biosciences).

## RNA extraction and RT-quantitative PCR

mRNA was extracted from cell lines and tissue samples using the TRIzol reagent (Invitrogen) or RNeasy Plus mini kit (Qiagen) according to the manufacturer's instructions. RNA (1 μg) was reverse transcribed with the MMLV reverse transcriptase (Invitrogen) or SuperScript IV VILO (Thermo Fisher Scientific). Quantitative PCR (qPCR) was performed with the SyBR Green Master Mix in a LightCycler 480 (Roche). Expression levels were normalized with the *36B4* or *GAPDH* human housekeeping genes. Primers used for qPCR are listed below.

| Target gene | Forward primer | Reverse primer |
|---|---|---|
| *GAPDH* | TCACCAGGGCTGCTTTTAAC | ATCTCGCTCCTGGAAGATGG |
| *36B4* | CATGCTCAACATCTCCCCCTTCTCC | GGGAAGGTGTAATCCGTCTCCACAG |
| *CD44* | CGCTTTGCAGGTGTATTCCA | ACCACGTGCCCTTCTATGAA |
| *Fra1* | CAGGCGGAGACTGACAAACTG | TCCTTCCGGGATTTTGCAGAT |
| *c-myc* | GTCAAGAGGCGAACACACAAC | TTGGACGGACAGGATGTATGC |
| *c-jun* | TCCAAGTGCCGAAAAAGGAAG | CGAGTTCTGAGCTTTCAAGGT |
| *CCND1* | CCCTCGGTGTCCTACTTCAA | AAGCGGTCCAGGTAGTTCAT |
| *LGR5* | CTCCCAGGTCTGGTGTGTTG | GAGGTCTAGGTAGGAGGTGAAG |
| *AXIN2* | TACACTCCTTATTGGGCGATCA | TTGGCTACTCGTAAAGTTTTGGT |
| *ASCL2* | CTCCCCACAGCTTCTCGACT | AGTGTCCCTCCAGCAGCTC |
| *SLC12A2* | TAAAGGAGTCGTGAAGTTTGGC | CTTGACCCACAATCCATGACA |

## Invasion and 3D cell migration assays

Cell invasion assays were performed in Fluoroblok invasion chambers (BD Bioscience) using 70,000–100,000 cells in the presence of 100 μl of 1–1.2 mg/ml Matrigel (BD Bioscience). After 24 h, cells were labelled with calcein AM (Sigma-Aldrich) and invasive cells were photographed using the EVOS FL Cell Imaging System. Quantification of the number of invasive cells per well was done with the ImageJ software. For 3D cell migration assays, 1,000 cells were resuspended in medium containing 2.4 mg/ml methylcellulose (Sigma-Aldrich) and plated in a U-bottom 96-well plate to allow the formation of one spheroid per well. Two days after, single spheroids were embedded in a 1:1 mix of 2.4 mg/ml neutralized bovine type I collagen (Purecol; Advanced BioMatrix) and medium supplemented with 12 mg/ml methylcellulose, according to the manufacturer's protocol. Phase-contrast photographs were taken daily. The invasive potential was determined by calculating the mean number of cells that moved further than an arbitrarily defined distance compared with the control condition set at 100%.

## Quantitative phosphoproteomics

Cells or mouse tumour samples were lysed in urea buffer (8M urea in 200 mM ammonium bicarbonate pH 7.5). Phosphopeptides were purified after tryptic digestion of 20 mg (for cells) or of 35 mg (for mouse tumours) total proteins using the PTMScan® Phospho-Tyrosine Rabbit mAb (PTyr-1000) Kit (Cell Signaling Technology), according to the manufacturer's protocol. An additional enrichment step using the IMAC-Select Affinity Gel (Sigma-Aldrich) was performed to increase the phosphopeptide enrichment. Purified phosphopeptides were resuspended in 10% formic acid, and two technical replicates for each sample were analysed using an EASY-nano LC system (Proxeon Biosystems, Odense, Denmark) coupled online with an LTQ-Orbitrap Elite mass spectrometer (Thermo Scientific, Waltham, MA). Each sample was loaded onto a 15-cm column packed in-house with 3 μM ReproSil-Pur C18 (75 μm inner

diameter). Buffer A consisted of $H_2O$ with 0.1% formic acid and buffer B of 100% acetonitrile with 0.1% formic acid. Peptides were separated using a gradient from 0 to 24% buffer B for 65 min, from 24 to 40% buffer B for 15 min and from 40 to 80% buffer B for 15 min (a total of 95 min at 250 nl/min). Data were acquired with the "Top 15 method", where every full MS scan was followed by 15 data-dependent scans on the 15 most intense ions from the parent scan. Full scans were performed in the Orbitrap at 120,000 resolution with target values of 1E6 ions and 500 ms injection time, while MS/MS ion trap scanning parameters were 1E4 ions as target value and 200 ms as maximum accumulation time. Database searches were performed with Mascot Server using the human Uniprot database (version 3.87). Mass tolerances were set at 10 ppm for the full MS scans and at 0.8 Da for MS/MS. Label-free quantification was performed on duplicate LC-MS runs for each sample using Progenesis LC-MS (Nonlinear Dynamics Software). Peptide intensities were normalized across all LC-MS runs, and normalized peptide intensities were summed for each unique phosphorylated peptide with a Mascot score exceeding 25. These intensities were then used to calculate the log2 fold change ratios of each unique phosphopeptide or each unique protein. In case of ambiguous phosphorylation site assignments, spectra were manually interpreted to confirm the localization of the phosphorylation site using Scaffold (Proteome software).

### In vivo experiments

In vivo experiments were performed in compliance with the French guidelines for experimental animal studies (Direction des Services Vétérinaires, Ministère de l'Agriculture, agreement B 34-172-27), and project authorizations following the 3R rules "Saisines no. 1,176–1,199 and no. 1,314–7,165" have been approved by the regional ethical committee. Housing was conducted according to Institutional Animal Care and local Committee guidelines. Five animals were allocated in each cage, and mice were housed at least 1 week to adapt to their new environment and avoid any stress bias before cell injection. Mice were randomized prior to treatment to determine random sampling such that the median luciferase signal between cohorts was the same. To minimize the effects of subjective bias, no information about groups and expected results was communicated to the experimentators. $2 \times 10^6$ cells were injected in the spleen of 5-week-old female athymic nude mice (Envigo). Two minutes after cell injection, spleen was removed to prevent the formation of primary tumours. Vehicle (0.5% methylcellulose/ 0.05% Tween 80) or nilotinib (50 mg/kg) was administered by oral gavage or i.p. daily, 5 days/week. After 4 weeks of treatment, liver was removed, photographed and cryopreserved. The metastatic index was determined by calculating the ratio between the metastatic nodule area and the total liver area. ctDNA level in the plasma of xenografted animals was measured using a quantitative PCR-based method that detects the K-RAS G13D and G12V point mutations present in human HCT116 and SW620 cells, respectively, as described in Thierry et al (2014) #251 and Mouliere, 2013 #252.

### Human tissue collection and transcriptomic analysis

All participating patients signed a written informed consent before enrolment, and the experiments conformed to the principles set out in the WMA Declaration of Helsinki and the Department of Health and Human Services Belmont Report. CRC specimens (metastases and histologically normal epithelium) ($n = 17$ patients) were obtained from the "Centre de Ressources Biologiques" (ICM, Montpellier, France) in accordance with the French government regulations and after approval by the local ethics committee. Proteins were extracted from frozen tissues at 4°C in lysis buffer using a Dual Glass Tissue Grinder size 21. Transcriptomic analysis was performed on a cohort of 143 stage IV CRC samples described in Del Rio et al (2017) #272. PFS was defined as the time from the beginning of the first-line metastatic treatment until recurrence or death. OS was calculated from the beginning of the first-line treatment until death. Survival was analysed using the Kaplan–Meier method, and differences between survival distributions were assessed using the log-rank and Breslow–Gehan tests.

### Luciferase reporter gene assays

10,000 HCT116 or HEK293 cells were seeded in 96-well plates 1 day before transfection. The next day, cells were transfected in triplicates with 60 ng of the TOPflash plasmid (TCF reporter) or of the mutant reporter FOPflash, 6 ng of a Renilla luciferase plasmid and 40 ng of β-catenin, DDR1 or DDR1 KD plasmids or empty vector, using Lipofectamine 2000 (Thermo Fisher Scientific). 48 hours after transfection, luciferase activity was quantified using the Dual-Glo Luciferase Assay System (Promega), according to the manufacturer' instructions, and the FLx800 microplate Fluorescence reader (BIO-TEK Instruments). If required, cells were incubated with 100 nM nilotinib overnight at ~32 h post-transfection.

### Data availability

The mass spectrometry proteomics data of the in vitro and in vivo analyses have been deposited to the ProteomeXchange Consortium via the PRIDE partner repository with the data set identifiers PXD008582 and PXD008546, respectively.

### Statistical analyses

All analyses were performed using GraphPad Prism. Data are presented as the mean ± SEM. When distribution was normal (assessed with the Shapiro–Wilk test), the two-tailed $t$-test was used for between-group comparisons. In the other cases, the Mann–Whitney test was used. Statistical analyses were performed on a minimum of three independent experiments. The statistical significance level is illustrated with P values: *$P \leq 0.05$, **$P \leq 0.01$, ***$P \leq 0.001$.

Expanded View for this article is available online.

### Acknowledgements

We gratefully thank C. Laschinger, N. Maurin and P. Mangeat for various reagents and plasmids; I. Ait Arsa and J. Vitre for technical assistance; and our colleagues for helpful discussions. We also thank M. Boyer (IGMM, Montpellier) for flow cytometry experiments, the Montpellier RIO Imaging and the RHEM histology platforms; P. Balaguer (Screening Platform in Cancer PCC, IRCM Montpellier) and M. Del Rio (IRCM, Montpellier) for the HCT116 LUC+ and SW620 LUC+ cell lines respectively, the Montpellier Genomic Collection (MGC), the Montpellier Platform of CRC preclinical models (MPCC), the "Centre de

**The paper explained**

**Problem**

Colon cancer (CRC) remains a lethal malignancy. While surgery can cure most of the patients with early stage CRC, only 10% of the patients diagnosed with a metastatic disease survive 5 years later. While novel therapies targeting protein kinases demonstrate some clinical benefit, they failed to significantly extend patient survival. Thus, there is an urgent need to identify novel therapeutic strategies in metastatic CRC. Since tyrosine kinases (TK) coordinate cell communication in mammals, their deregulation may play a major role in this process.

**Results**

We uncovered a central metastatic role for the TK activity of receptor for collagens DDR1 in CRC. Our proteomic analyses support a model where sustained DDR1 activation upon collagen deposition at the tumour front or at the metastatic niche maintains a high level of β-catenin transcriptional activity necessary for cancer stem cells dissemination and survival. Consistent with this idea, pharmacological DDR1 inhibition with nilotinib, a TK inhibitor currently used in the clinic to target the BCR-ABL oncogene in chronic myeloid leukaemia, displays a potent anti-metastatic activity in CRC.

**Impact**

Our findings may lead to the design of novel therapeutic strategies in advanced CRC, which is urgently needed to cure CRC patients. Specifically, it may lead to the repositioning of a clinically approved drug in leukaemia for metastatic CRC, which can be quickly implemented in the clinic.

Ressources Biologiques" (ICM, Montpellier) and the Montpellier Small Animal Imaging Platform (IPAM) of Montpellier. This work was supported by ARC, Montpellier SIRIC Grant "INCa-DGOS-Inserm 6045", Ligue Nationale contre le Cancer, CNRS and the University of Montpellier. CL and MJ were supported by the Ligue Nationale contre le Cancer; PT, JLG and LCT by the Montpellier SIRIC; ML by the Fondation pour la Recherche Médicale; VS and JP by the CNRS; BM, PM, ART and SR by the INSERM; and AS by the Fondation de France and the CNRS.

## Author contributions

All authors contributed extensively to the work presented in this article. Experimental analysis and data acquisition: MJ, CL, PT, ML, JLG, VS, DB, BR, CM, SEM, AO, LC-T, MB, ART, PM, JP and AS. Contribution with tools: FG and VS. Writing of the article: MJ, AS and SR. Project supervision: SR and AS.

## Conflict of interest

The authors declare that they have no conflict of interest.

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
