## [Review Process File · EMBO Molecular Medicine]

Inhibition of DDR1-BCR signaling by nilotinib as a new therapeutic strategy for metastatic colorectal cancer

Maya Jeitany, Cedric Leroy, Priscillia Tosti, Marie Lafitte, Jordy Le Guet, Valérie Simon, Debora Bonenfant, Bruno Robert, Fanny Grillet, Caroline Mollevi, Safia El Messaoudi, Amaëlle Otandault, Lucile Canterel-Thouennon, Muriel Busson, Alain R. Thierry, Pierre Martineau, Julie Pannequin, Serge Roche, Audrey Sirvent

Review timeline:

Submission date:	19 April 2017
Editorial Decision:	19 May 2017
Additional correspondence (author)	30 May 2017
Additional correspondence (author)	12 September 2017
Additional correspondence (editor)	15 September 2017
Revision received:	16 November 2017
Editorial Decision:	13 December 2017
Revision received:	10 January 2017
Accepted:	15 January 2017

Editors: Roberto Buccione and Céline Carret

Transaction Report:

1st Editorial Decision

19 May 2017

Thank you for the submission of your manuscript to EMBO Molecular Medicine. We are very sorry that it has taken so long to get back to you on your manuscript. Unfortunately, we experienced some difficulties in securing three willing and appropriate reviewers and then obtaining their evaluations in a timely manner.

As you will see, the three Reviewers find the study of significant interest and worthy of publication but also raise a number of relevant and partially overlapping concerns. I will not go into much detail but I would like to highlight a few: need for better support for the suggested mechanism of Wnt pathway activation downstream of Collagen-DDR1; lack of direct demonstration the RAS independent role of DDR1 in mediating migration of CRC; unclear whether DDR1 inhibition by nilotinib is equally effective in the context of RAS/BRAF wild-type colorectal cancer cells and need for multivariate analysis to rule out confounding factors in suggesting the prognostic value of DDR1 expression levels.

I would also like to mention that there is clearly a requirement to directly show the potential efficacy of nilotinib on established metastatic lesions to enhance the translational/clinical relevance of the manuscript (of great relevance for EMBO Molecular Medicine)

During our reviewer cross-commenting exercise, a consensus clearly emerged on the fundamental importance of the experimentation on established metastatic lesions and also the need to better support the prognostic value of DDR1 with multivariate analysis.

We are thus prepared to consider a substantially revised submission, with the understanding that the Reviewers' concerns must be addressed with additional experimental data where appropriate and

that acceptance of the manuscript will entail a second round of review. In agreement with reviewer 3, we will not be asking you to investigate the role of the tumour microenvironment, but we would of course welcome additional data if available.

Since the required revision in this case appears to require a significant amount of time, additional work and experimentation and might be technically challenging, I would therefore understand if you chose to rather seek publication elsewhere at this stage. Should you do so and although we hope not, we would appreciate a message to this effect. Please note that it is EMBO Molecular Medicine policy to allow a single round of revision only and that, therefore, acceptance or rejection of the manuscript will depend on the completeness of your responses included in the next, final version of the manuscript.

I look forward to seeing a revised form of your manuscript in due time.

***** Reviewer's comments *****

Referee #1 (Comments on Novelty/Model System):

Experiments have been performed in several cell lines and primary tumor models therefore excluding cell line specific effects. In vitro functional data for DDR1 inhibition have been verified in xenograft mouse models of CRC growth and metastasis to the liver. However, the analyses of BCR functional contribution to DDR1 inhibition is not complete and does not justify the conclusions made by the authors in my opinion.

Biostatistics, according to the methods section, are based on a sufficient number of independent experiments.

The finding that Nilotinib via DDR1 inhibition can be used to prevent CRC migration and metastasis is a novel finding of high medical impact.

The used model systems are adequate. However, the functional role of BCR in metastasis has been addressed exclusively in ex vivo models which represents a weakness of the manuscript. This issue has been addressed in more detail in the remarks to be sent to the authors.

Referee #1 (Remarks):

In their manuscript entitled „Inhibition of DDR1-BCR signalling by nilotinib as a new therapeutic strategy for metastatic colorectal cancer“, Jeitany M. and colleagues describe the collagen receptor DDR1 as a druggable target in CRC disease. Inhibition of DDR1 receptor tyrosine kinase activity by nilotinib blocks CRC cell migration and metastasis, and the downstream DDR1 substrate BCR is important for CRC cell migration ex vivo. Furthermore, DDR1 shows increased kinase activity in CRC liver metastatic specimen and DDR1 expression predicts shorter overall survival in CRC patients.

While DDR1 pharmacological inhibition has been described for other cancer entities, the here presented manuscript extends this concept to colorectal cancer, independent of its KRAS mutational status. Hence, the findings described by Jeitany M. et al. are of broad interest to the research field. This in principle allows publication in a renowned journal such as EMBO Molecular Medicine. Mechanistically, the authors suggest that phosphorylation of BCR by DDR1 disrupts its interaction with β -catenin which then can localize to the nucleus to transactivate pro-migratory Wnt target genes. However, the authors do not provide sufficient experimental evidence that would support this hypothesis, and the shown experimental data on this topic are difficult to interpret. More experiments are needed in order to clarify this issue.

Major points:

1. Notably, no functional data from in vivo experiments have been presented by the authors that would support a direct role of phospho-BCR in CRC liver metastasis. While Fig.5 shows functional evidence for a role of phospho-BCR in ex vivo cell migration and invasion assays, the study lacks xeno-engraftment experiments using CRC cells with BCR knock-down and/or expression of the Y177F BCR mutant that would demonstrate the importance of phospho-BCR in this scenario. The authors should emphasize that the contribution of BCR phosphorylation downstream of DDR1 to CRC liver metastasis needs further clarification. Also other DDR1 substrates might play a role in this process.
2. In Figure 5, the authors demonstrate that mere DDR1 over-expression in cells does not lead to a significant BCR phosphorylation in the absence of collagen. However, results shown in Figure 6

suggest that DDR1 ectopic expression alone is sufficient to drive Wnt target gene induction in SW620 cells. It would be important to address whether collagen further increases the effect of DDR1 on Wnt activity in order to support a role of Phospho-BCR in this scenario.

3. While phosphorylation status-dependent interaction of BCR with β -catenin has been shown previously by Renshaw & Moelling, as cited correctly in the manuscript, the authors do not provide any evidence that the demonstrated increase in Fra1, c-MYC, and Jun mRNA level upon DDR1 ectopic expression is dependent on this mechanism. A knock-down of endogenous BCR + ectopic expression of BCR Y177F should prevent activation of Wnt signaling upon DDR1 activation. Since BCR Y177F does not mediate cell invasion of HCT116 and SW620 cells, as shown in Figure 5H, it should still be able to sequester B-catenin to the cytoplasm in this setting if the mechanism suggested by the authors is correct.

4. Intriguingly, knock-down of BCR reduces cell migration. If less BCR is available to sequester B-catenin to the cytoplasm, one would expect a similar effect as suggested for BCR-phosphorylation: a stronger activation of Wnt signaling and hence more cell migration. The opposite effect is shown in the manuscript. Therefore, phosphorylation-independent effects of BCR on cell migration seem more likely.

5. Localization of B-catenin should be quantified in HCT116 and SW620 cells via nuclear/cytoplasmic fractionation upon BCR knock-down, ectopic expression, or BCR Y177F ectopic expression in order to support the here suggested mechanism of Wnt pathway activation downstream of Collagen-DDR1.

6. The authors should also look at other canonical Wnt target genes, such as Lgr5, Axin2, Ascl2, and Slc12a2. If and how their expression is modulated by the Collagen-DDR1-BCR axis should be analyzed in more detail.

Minor points:

1. Colonization of the liver by CRC cells and maintenance of liver metastatic CRC tumor growth has been shown to depend on an LGR5+ tumor cell population as has been recently shown by the laboratory of Frederic Sauvage (Melo FS et al., Nature, 2017). Since the authors have demonstrated that treatment with nilotinib prevented formation and progression of already established liver metastatic nodules (Fig. 3), it would be very interesting to address whether nilotinib or inactivation of DDR1 by siRNA-mediated knock-down reduces the level of LGR5 expression in this setting.

Referee #2 (Comments on Novelty/Model System):

My reservations on the medical impact are due to the fact that the authors have not clearly addressed whether their therapeutic strategy will be effective in established metastatic lesions. This is discussed in the comments to the authors.

Referee #2 (Remarks):

Manuscript EMM-2017-07918

The metastatic dissemination of colorectal (CRC) cancer, most frequently to liver and lung, is the main cause of mortality in these patients. The manuscript by Jeitany and co-workers describes a novel strategy to diminish the invasiveness of CRC cell lines in vitro and liver metastasis following intrasplenic injection in nude mice.

In this manuscript, the authors have used several human CRC cell lines to evaluate the anti-metastatic capacity of the clinically approved TKI nilotinib. Although the rationale for selecting this drug is not clearly discussed, the authors convincingly demonstrate that the anti-metastatic capacity of nilotinib is mediated by the inhibition of DDR1. This is achieved using a combination of phosphoproteomics and several in vitro and in vivo approaches. Furthermore, the current study identified BCR as a novel DDR1 substrate. BCR has been previously reported to function as a negative regulator of beta-catenin in CRC. Jeitany and colleagues now show that DDR1-mediated phosphorylation on Tyr177 inactivates BCR resulting in increased expression of beta-catenin transcriptional targets.

One of the concerns typically associated to studies using imatinib derivatives is their broad specificity and therefore the difficulty to validate the importance of individual drug-targets. However, the authors have carried out elegant gain and loss of function experiments (in particular reconstitution experiments with the nilotinib resistant DDR1 T701I mutant) to convincingly

demonstrate the central role of DDR1 inhibition in mediating the nilotinib response. This was particularly important to rule out the implication of ABL as the kinase responsible for the Tyr177 phosphorylation of BCR.

Finally, the authors describe that DDR1 overexpression is associated with shorter progression-free and overall survival in CRC patients and that DDR1 activity might be increased in metastatic compared to primary lesions. As such, the authors propose that DDR1 inhibition could be effective in patients with metastatic CRC.

In sum, the research described here was excellently conceived, executed and presented. Yet I feel that additional evidence should be incorporated before the manuscript is ready for publication.

Major points:

1. Central to their proposed model (see Figure 8) is the role of beta-catenin downstream of DDR1. In brief, BCR interacts with beta-catenin in the cytoplasm to prevent its nuclear localization resulting in repression of its target genes. In this model DDR1-dependent BCR Tyr177 phosphorylation disrupts this interaction thereby relieving beta-catenin inhibition.

To prove this and to evaluate the impact of nilotinib and DDR1 in beta-catenin function the authors have used a beta-catenin reporter plasmid (Fig 6A, B) and have also assessed the levels of various transcriptional targets (Fig 6C). Yet, given the central role in their model they should also provide experimental evidence showing that both nilotinib treatment and DDR1 depletion increase the pool of nuclear beta-catenin. This would be particularly informative in the liver metastasis model described in figure 3F. Immunohistochemistry staining of liver sections to evaluate the subcellular localization of beta-catenin following nilotinib treatment and DDR1 gain/or loss of function experiments would reinforce their proposed hypothesis.

2. The concept of DDR1 mediating a RAS-independent mechanism in CRC is profusely used during the manuscript. Yet, in my opinion the authors have not proved that this is the case. It is true that they demonstrate that DDR1 activation (page 9) or nilotinib treatment (page 10) fails to affect MAPK or AKT phosphorylation. Yet, this experiment shows that DDR1 is not essential for the activity of these two pathways downstream of RAS, but by no means rules out the possibility that DDR1 function requires RAS and/or is somehow regulated by RAS activity. Indeed, 9 out of the 11 CRC cell lines in which the authors have assessed the effect of nilotinib inhibition on cell invasion (see Figure 1) have either KRAS or BRAF oncogenic mutations; only SW48 and Caco2 are wild-type for both. Incidentally, one of the two in which nilotinib has no effect is indeed SW48.

In order to prove the RAS independent role of DDR1 in mediating migration of CRC cells the authors would require inhibiting KRAS function and evaluate the impact on DDR1 activity. This is not an easy experiment but at least the authors could assess the effect of MEK inhibition in the context of collagen-activated DDR1. For instance, is BCR phosphorylation on Tyr177 following collagen treatment reduced in the presence of MEK inhibitors?

In the absence of this result the authors should be cautious when referring to the putative RAS-independent therapeutic strategy.

3. The authors suggest that nilotinib treatment might also affect the growth of established metastasis. In support of this hypothesis the authors provide the following data: "nilotinib treatment prevented liver metastatic progression, as confirmed by the significant decrease of ctDNA level compared with DMSO treated animals (Fig 3F)". While this could be true, it is also possible that in this context nilotinib might be inhibiting the release of CRC cells from the primary spleen injection from day 7 onwards (that is when nilotinib treatment starts). There is a much more direct way of assessing the effect on metastasis growth. The cell line used in this experiment to perform the intrasplenic injection is luciferase positive. Indeed, the authors only start the nilotinib treatment "when luciferase-positive metastases were already detectable" (see page 8). Being this the case, the authors could monitor metastasis growth during the course of the two-week nilotinib treatment by also using luciferase signal as a surrogate marker of liver tumour burden.

Minor points:

a. Given the difficulty to access patient samples it would have been ideal to study the phosphorylation of BCR Tyr177 in the patient samples primary/metastasis shown in Figure 7B and

to evaluate its correlation with DDR1 activity.

Referee #3 (Comments on Novelty/Model System):

This is a solid paper which conveys novel information in the field of colorectal cancer translational research. The introduction provides extensive relevant background information (some of which can be shortened), but the objectives of the work could be better spelt out. The results are presented in a logical workflow, from descriptive data in preclinical models, to mechanistic insights, then relevance in human CRC with studies on patient samples (including *in vitro* experiments on patient derived models). The results are accurately described and the discussion is sound with pertinent references to previous works. The experimental procedures have been clearly reported with sufficient details to allow replication.

I believe this work could be of interest to the readers of EMBO molecular medicine, after addressing major points #1 (need to use not mutated cell models), #2 and #3 (multivariate statistical analysis should be performed) that I have outlined in the authors' comments. Experiments to address point #4 may be considered beyond the scope of this work (which is already quite extensive), but I suggest the point should at least be discussed.

Referee #3 (Remarks):

This elegant work reports that nilotinib can inhibit colorectal cancer cell invasion *in vitro*, and it can also reduce liver metastasis formation when RAS mutant colon cancer cells are inoculated intrasplenically in mouse models. Nilotinib impairs invasion by inhibiting collagen I dependent phosphorylation of DDR1 in CRC cells (including lines derived from patient circulating tumor cells). Mechanistically, nilotinib prevents DDR1-mediated BCR phosphorylation on Tyr177, which, in turn, is important for maintaining beta-catenin transcriptional activity necessary for cancer cell invasion. Phosphorylation of DDR1 increases over tumor progression stages, and DDR1 expression is apparently correlated with poor prognosis in advanced CRC patients.

The study is nicely presented and it describes some potentially novel and relevant information for the field of translational colorectal cancer research. However, a number of points should be clarified or extended.

Major Points

1) Most of the *in vitro* invasion assays and all the *in vivo* experiments have been performed using RAS mutated (HCT116, SW620 and CPP19) or BRAF mutated (HT29, CPP30, CTC44, CTC45) tumour cells. It is not clear whether DDR1 inhibition by nilotinib is equally effective in the context of RAS/BRAF wild-type colorectal cancer cells. For instance, nilotinib is able to reduce liver metastasis formation of HCT116 cells injected intrasplenically (Fig. 1D). This experiment should be replicated in at least an independent RAS/BRAF wild type cell line and, if possible, also using a RAS/BRAF mutant patient CTC derived cell line.

2) Does nilotinib affect viability or proliferation of CRC cells? The answer is most likely no, but data should be provided and this point should be clearly mentioned within the text.

3) Expression level of DDR1 seems associated with shorter overall survival in patients with metastatic CRC. When establishing the prognostic value of a novel marker, multivariate analyses should be performed to rule out the confounding effects of any other variables known to be associated with survival. In the metastatic setting, several baseline variables are known to affect prognosis, including, RAS/BRAF mutational status, tumor location (right vs left), MSI-high status, ECOG performance status, mucinous histology, primary resection, time to metastasis (metachronous vs synchronous mets), number of metastatic sites, transcriptional subtypes. Are any of these variables associated with higher expression levels of DDR1?

4) All experiments have been performed on the cancer cell compartment, while an effect of nilotinib on tumor-tumor microenvironment interaction should also be taken into account. DDR1 is known to

induce extracellular matrix remodeling. It would be interesting to learn whether and how nilotinib treatment affects how colon cancer cells interact with their tumor microenvironment. For instance, the authors could investigate whether nilotinib is able to inhibit the interaction/adhesion of HCT116 tumour cells with cell types representative of the microenvironment, such as human endothelial cells, pericytes, fibroblasts and hepatocytes in 3D co-culture systems, as well as its effect on remodeling of the extracellular matrix (are specific metalloproteinases modulated by nilotinib in CRC cells?).

Minor Points

5) Nilotinib does not impair migration of SW48 or DLD1 cells, and only minimally affects CTC45. Can the authors speculate about explanations underlying this cell type specific effect?

6) The introduction is overtly long and could be shortened by providing only the essential information about metastatic colorectal cancer (relatively) poor prognosis, and the use of TK inhibitors in CML (pages 3-4).

7) Page 6, first paragraph. It is not clear why the authors have chosen to analyze a KRAS mutant cell line (HCT116) 'To search for RAS-independent therapeutic strategies for metastatic CRC'. I would rephrase the paragraph saying that nilotinib displays anti-invasive activity in a panel of CRC cell lines, irrespective of their genotype. Indeed, Figure 1C can be improved by providing the genotype below the name of CRC cell lines in order to make it more immediate that nilotinib can reduce cell invasion independent of the tumor mutational status (if this is the case).

8) I invite the authors to speculate and discuss on possible combinations of nilotinib with agents targeting the Wnt-Beta-catenin pathway.

9) Representative pictures of cell line invasion assays used for the quantification histograms shown in Figure 1C should be provided as supplemental information.

Additional correspondence - author

30 May 2017

We are very pleased to hear that EMM and the reviewers are very positive about our ms.

After discussing with our collaborators, we plan to address most reviewers' concerns and send a revised version of our ms asap.

Now, we feel that the most challenging and time consuming aspects of raised concerns are the in vivo experiments:

we plan to test metastatic abilities of our DDR1-SW620 and/or HCT116 BCR cell derivatives in nude mice to address the in vivo role of pTyr177-BCR in DDR1 signaling, as requested by the reviewer #1.

-We also plan to test the activity of nilotinib on metastatic abilities of KRAS/BRAF WT CRC cells as requested by the reviewer #3. However we are facing an additional issue here, ie we could not find any KRAS/BRAF WT CRC cell-lines known to induce metastatic nodules when spleen-injected in nude mice.

To solve this problem, we are currently testing the KRAS/BRAF status of primary cultures derived from metastatic CRC patients that were developed by one of our collaborator (CCP cells). We just realized that CPP30 cells are BRAF/KRAS WT (and not mutated for BRAF as originally reported). Whether these cells are metastatic in nude mice is however currently unknown. Thus, we will test their metastatic ability in nude mice and the nilotinib activity in this KRAS WT model. In the mean time, we are screening for additional KRAS/BRAF wt CPPs as a backup model.

Practically, it will be very difficult to perform all these experiments within the next 3 months, specifically during summer time.

Therefore, we would be more than happy if we could get at least one month extension for the reviewing of our ms. Anyway, I will keep you inform about the advance of our reviewing experiments on time.

I would like to thank you again for considering our ms for publication in EMM.

(Editor agreed to extension.)

Additional correspondence - author

12 September 2017

We would like to inform you on our progress to answer reviewers' concerns. Here is summarized the data we have obtained so far:

Referee #1

We have now strong mechanistic evidence for a DDR1-BCR-beta-catenin signaling operating in CRC cells in vitro. Specifically, we show by imaging methods that collagen stimulation of DDR1-SW620 cells induces accumulation of active beta-catenin in the nucleus. This molecular response is regulated by the kinase-activity of DDR1 (nilotinib inhibition) and phosphorylation of BCR on Tyr177.

We have functional evidence further supporting our model, ie BCR is a negative regulator of cell migration, which is alleviated by phosphorylation on Tyr177 upon collagen stimulation. Specifically, we show that collagen increases cell invasion and BCR phosphorylation on tyr177. Consequently, BCR depletion inhibits cell invasion in the presence of collagen. However BCR is not phosphorylated in the absence of collagen and in this conditions, BCR depletion increases cell migration in Boyden chamber.

We now show that DDR1 depletion inhibits additional canonical Wnt target genes, ie Cyclin D1, CD44, Lgr5 and Axin2. However it has no effect on Ascl2 and Slc12a2, suggesting a specific effect of DDR1 signaling on a sub-group of Wnt target genes.

Referee #2

We provide further evidence for the Ras-independent nature of DDR1-BCR signaling in CRC cells. Specifically, we show that a MEK inhibitor has no effect on this pathway.

We have data on Luciferase signal as a surrogate marker of liver tumor burden.

Referee #3

We now show that patient CPP30 cells are WT for RAS and BRAF. We then used this cellular model to replicate the anti-metastatic activity of nilotinib in an additional independent RAS/BRAF wild-type cell-line. We have now in vivo evidence showing that nilotinib treatment significantly reduces the liver metastatic burden in nude mice that have been inoculated with these patient-derived tumor cells.

We have now data showing that nilotinib poorly affects the standard proliferation of CRC cells in vitro (2D).

We have performed univariate analysis from our CRC samples. We now show a correlation between survival (progression free survival or overall survival) and DDR1 expression, WHO performance status, tumor location, grade, number of metastatic sites, MSI, and the molecular subtype of the primary tumor (CMS). Using multivariate regression analysis, DDR1 expression remains significantly associated with the progression-free survival, together with number of metastatic sites. Some known factors like BRAF/RAS status were not significant in our test, because a too small fraction of the patients has been analyzed (only 33/143). None of these variables are associated with

DDR1 expression; in particular, there is no significant difference in the expression level of DDR1 between the four CMS subtypes.

However there are still some data we could not obtain so far:

Referee #1

We still did not obtain functional evidence for a role of BCR Tyr177 in metastatic formation induced by DDR1 in nude mice. We have performed two attempts by spleen-injecting DDR1-SW620 cells depleted for BCR (shBCR) and re-expressing either BCR WT or the Y177F mutant in recipient animals. We have performed an additional attempt with HCT116 cells expressing shBCR. However, we had lost shBCR expression in tumors from several animals, probably because BCR is central for these CRC cells transforming properties. Accordingly, we loose shBCR expression in these CRC cells also in vitro after 2-3 passages in culture.

Referee #2

We still do not yet have IHC data of liver sections showing the subcellular localization of beta-catenin following nilotinib treatment and DDR1 gain/or loss of function experiments; however we are currently setting up the conditions suitable for IHC analysis.

In summary, we think we can answer most reviewers' concerns. Specifically we think we have now strong mechanistic evidence for a collagen-DDR1-BCR-beta-catenin signaling operating in CRC cells in vitro. We have further evidence for the RAS-independent nature of DDR1 signaling in these tumor cells. We could confirm the anti-metastatic activity of nilotinib in an additional CRC model WT for RAS and BRAF close to the human pathology. We think that this piece of data will give a strong credit to the interest of nilotinib in anti-cancer therapy in CRC.

Regarding the effect of DDR1-BCR signaling on beta-catenin activity in vivo, we propose to perform IHC analyses from our experimental metastatic tumors in mice to get an in vivo correlation between DDR1 expression, BCR phosphorylation and nuclear beta-catenin activity. We may also be in a position to address the in vivo effect of nilotinib on these markers. We think this data should get additional support to the proposed DDR1 metastatic pathway in CRC. Practically, we would ask for a 2 months extension to obtain this data.

Regarding the functional role of BCR phosphorylation in vivo, we are ready to make a fourth attempt using new freshly made DDR1-SW620 cells KD for BCR and re-expressing either BCR WT or the YF mutant and inject them in NOD SCID mice to favor metastatic induction and reduce clonal selection. However, to be franc and honest, we are not very positive about the success of this experiment. We think that a CRISPR strategy to KO BCR may not very informative either as we may use a specific cellular clone, unless we inject the whole cell population. However in those conditions, we may still obtain liver metastases from non BCR KO cellular clones. Practically, we think this set of experiments may require an additional 4 months from now.

As an alternative, we would suggest to address the in vitro role of the DDR1-BCR pathway on CRC cells growth in the presence of collagen. This in vitro assay would support a growth-promoting role of DDR1-BCR signaling in conditions close to the in vivo situation (collagen-enriched metastatic niche). This idea is supported by the recent results of Gao et al (Cell 2016), who described a 3D cell culture condition to address the role of collagen on breast tumor cells metastatic activation. We think that these piece of data together with in vivo correlation between DDR1 expression, BCR phosphorylation and beta-catenin activity in experimental metastatic tumors would further support our proposed DDR1 metastatic pathway in CRC.

Please let us know what you think about all this and what we should do in the hope our manuscript to be suitable for publication in EMM.

Additional correspondence - editor

15 September 2017

Thank you very much for this update and for expressing your concerns and plan. [...]

That said, I believe that what you propose is sound and in particular I would not encourage another *in vivo* attempt with the modified DDR1-SW620 cells. The alternative you propose in this respect appears reasonable.

Although I cannot speak for the reviewer(s), as long as you clearly explain, just as you do here, the reasons for your experimental strategies in the rebuttal, I foresee no significant issues. However, I can directly contact the reviewer on this specific aspect if you wish me to.

I hope this helps.

1st Revision - authors' response

16 November 2017

Referee #1 (Comments on Novelty/Model System):

Experiments have been performed in several cell lines and primary tumor models therefore excluding cell line specific effects. In vitro functional data for DDR1 inhibition have been verified in xenograft mouse models of CRC growth and metastasis to the liver. However, the analyses of BCR functional contribution to DDR1 inhibition is not complete and does not justify the conclusions made by the authors in my opinion.

Biostatistics, according to the methods section, are based on a sufficient number of independent experiments.

The finding that Nilotinib via DDR1 inhibition can be used to prevent CRC migration and metastasis is a novel finding of high medical impact.

The used model systems are adequate. However, the functional role of BCR in metastasis has been addressed exclusively in ex vivo models which represents a weakness of the manuscript. This issue has been addressed in more detail in the remarks to be sent to the authors.

Referee #1 (Remarks):

In their manuscript entitled "Inhibition of DDR1-BCR signalling by nilotinib as a new therapeutic strategy for metastatic colorectal cancer", Jeitany M. and colleagues describe the collagen receptor DDR1 as a druggable target in CRC disease. Inhibition of DDR1 receptor tyrosine kinase activity by nilotinib blocks CRC cell migration and metastasis, and the downstream DDR1 substrate BCR is important for CRC cell migration ex vivo. Furthermore, DDR1 shows increased kinase activity in CRC liver metastatic specimen and DDR1 expression predicts shorter overall survival in CRC patients.

While DDR1 pharmacological inhibition has been described for other cancer entities, the here presented manuscript extends this concept to colorectal cancer, independent of its KRAS mutational status. Hence, the findings described by Jeitany M. et al. are of broad interest to the research field. This in principle allows publication in a renowned journal such as EMBO Molecular Medicine. Mechanistically, the authors suggest that phosphorylation of BCR by DDR1 disrupts its interaction with beta-catenin which then can localize to the nucleus to transactivate pro-migratory Wnt target genes. However, the authors do not provide sufficient experimental evidence that would support this hypothesis, and the shown experimental data on this topic are difficult to interpret. More experiments are needed in order to clarify this issue.

We thank the reviewer for his/her positive comments. We feel we have now obtained additional data supporting our hypothesis (see below).

Major points:

1. Notably, no functional data from in vivo experiments have been presented by the authors that would support a direct role of phospho-BCR in CRC liver metastasis. While Fig.5 shows functional evidence for a role of phospho-BCR in ex vivo cell migration and invasion assays, the study lacks xeno-engraftment experiments using CRC cells with BCR knock-down and/or expression of the Y177F BCR mutant that would demonstrate the importance of phospho-BCR in this scenario. The

authors should emphasize that the contribution of BCR phosphorylation downstream of DDR1 to CRC liver metastasis needs further clarification. Also other DDR1 substrates might play a role in this process.

We have performed two attempts by spleen-injecting DDR1-SW620 cells depleted for BCR (shBCR) and re-expressing either BCR WT or the Y177F mutant in recipient animals. We have performed an additional attempt with HCT116 cells expressing shBCR. However, shBCR expression had been lost in tumors from several animals, probably because BCR is central for these CRC cells transforming properties. Accordingly, we lose shBCR expression in these CRC cells also in vitro after 4-5 passages in culture. We think that a CRISPR strategy to KO BCR may not be very informative as we may use a specific cellular clone, unless we inject the whole cell population. However in those conditions, we may still obtain liver metastases from non BCR KO cellular clones. As an alternative, we have attempted to address the in vitro role of the DDR1-BCR pathway on the capacity of CRC cells to form oncospheres in the presence of collagen. This idea is supported by the recent results of Gao et al (Cell 2016), who described a 3D cell culture condition to address the role of collagen on breast metastatic tumour cells activation. This in vitro assay supports a growth-promoting role of DDR1 signalling in conditions closer to the in vivo situation (collagen-enriched metastatic niche) and outlines a partial role of DDR1 TK activity in this process. However, BCR was not involved in this process, highlighting the implication of additional substrate-dependent signaling pathways (Referees Fig 1). Now, these set of experiments cannot lead to a definite conclusion as it is possible that this assay does not recapitulate the level (and nature of reticulation) of collagen deposition at the metastatic niche, which may dictate a distinct TK-dependent role of DDR1 on metastatic growth.

Referees Figure 1. DDR1 affects colospheres growth in collagen I but this effect is only partially dependent of DDR1 kinase activity and of BCR.

Nevertheless, we have obtained additional experimental evidence supporting the role of BCR on β -catenin signalling, specifically on nuclear active β -catenin accumulation induced by DDR1 activation (see major point 2 and new Fig 6E). Consistent with our model, IHC analysis shows that DDR1 expression significantly increases the level of active β -catenin in the nucleus of metastatic tumors (new Fig EV5C), while nilotinib treatment reduces this molecular effect (new Fig 6F). In conclusion, we feel that this additional results further support our hypothesis.

We have incorporated a comment on additional oncogenic pathways activated by DDR1 in the revised discussion (p17).

2. In Figure 5, the authors demonstrate that mere DDR1 over-expression in cells does not lead to a significant BCR phosphorylation in the absence of collagen. However, results shown in Figure 6 suggest that DDR1 ectopic expression alone is sufficient to drive Wnt target gene induction in SW620 cells. It would be important to address whether collagen further increases the effect of DDR1 on Wnt activity in order to support a role of Phospho-BCR in this scenario.

We do see a significant BCR phosphorylation upon DDR1 overexpression, even in the absence of collagen I. Specifically, the western-blot incorporated in Fig 5C has been replaced by a western-blot of better quality and of longer exposure that clearly supports our conclusion. Consistent with this idea, we have now additional and strong mechanistic evidence for a DDR1-BCR- β -catenin signaling operating in CRC cells in vitro. Specifically, we show by imaging methods, that DDR1 overexpression alone in SW620 induces accumulation of active β -catenin in the nucleus (new Fig EV5B). This β -catenin activity is further increased by collagen I stimulation. Importantly, this molecular response is regulated by the kinase-activity of DDR1 (nilotinib inhibition) and phosphorylation of BCR on Tyr177 (prevented by re-expression of BCR YF mutant) (new Fig 6E). In addition, we performed IHC on mouse liver sections. We observed a correlation between collagen I accumulation and the intensity of active β -catenin immunostaining at the edge of the metastases (Referees Figure 2).

SW620 DDR1 liver metastasis

Referees Figure 2. Correlation between collagen I and active β -catenin immunostaining in SW620 DDR1 derived liver metastasis.

3. While phosphorylation status-dependent interaction of BCR with beta-catenin has been shown previously by Ressa & Moelling, as cited correctly in the manuscript, the authors do not provide any evidence that the demonstrated increase in *Fra1*, *c-MYC*, and *Jun* mRNA level upon DDR1 ectopic expression is dependent on this mechanism. A knock-down of endogenous BCR + ectopic expression of BCR Y177F should prevent activation of Wnt signaling upon DDR1 activation. Since BCR Y177F does not mediate cell invasion of HCT116 and SW620 cells, as shown in Figure 5H, it should still be able to sequester B-catenin to the cytoplasm in this setting if the mechanism suggested by the authors is correct.

Our imaging data demonstrate that collagen stimulation of DDR1-SW620 cells induces accumulation of active β -catenin in the nucleus and that this molecular response is regulated by phosphorylation of BCR on Tyr177 as cells expressing BCR Y177F mutant affects the capacity of DDR1 to promote the active beta-catenin accumulation in the nucleus (new Fig 6E).

4. Intriguingly, knock-down of BCR reduces cell migration. If less BCR is available to sequester B-catenin to the cytoplasm, one would expect a similar effect as suggested for BCR-phosphorylation: a stronger activation of Wnt signaling and hence more cell migration. The opposite effect is shown in the manuscript. Therefore, phosphorylation-independent effects of BCR on cell migration seem more likely.

We have functional evidence further supporting our model, ie BCR is a negative regulator of cell migration, which is alleviated by phosphorylation on Tyr177 upon collagen stimulation. Specifically, we show that when the level of pTyr177 BCR in CRC cells was low in the absence of collagen (absence of matrigel), BCR depletion increased cell migration in Boyden chamber assays (new Fig EV6). This effect was abolished by BCR and BCR Y177F expression, highlighting the phosphorylation-independent BCR anti-migratory activity. In contrast, when BCR phosphorylation on Tyr177 was increased upon collagen stimulation (collagen IV present in matrigel) and its depletion reduced CRC cell invasion in matrigel (Fig 5G & H). Invasion was restored by BCR but not BCR Y177F expression, highlighting the phosphorylation-dependent BCR invasive activity. A comment on this point has been added in the results part (p13).

5. Localization of B-catenin should be quantified in HCT116 and SW620 cells via nuclear/cytoplasmic fractionation upon BCR knock-down, ectopic expression, or BCR Y177F ectopic expression in order to support the here suggested mechanism of Wnt pathway activation downstream of Collagen-DDR1.

This point has been addressed by an imaging method in SW620 cells, which we feel is more accurate than fractionation assays, as it prevents any bias due to recurrent contamination by ER proteins in nuclear fractions (new Fig 6E and see above the answer to the major point 3).

6. The authors should also look at other canonical Wnt target genes, such as Lgr5, Axin2, Ascl2, and Slc12a2. If and how their expression is modulated by the Collagen-DDR1-BCR axis should be analyzed in more detail.

We have analyzed the role of DDR1 on additional canonical Wnt target genes (transcript level) in HCT116 cells. We confirmed that DDR1 expression regulates the expression of Wnt target genes CD44, CCND1, LGR5 and AXIN2 but it has no effect on ASCL2 and SLC12A2 (new Fig EV5A). Collagen-DDR1-BCR axis on β -catenin signaling has been addressed by imaging methods (new Fig 6E).

Minor points:

1. Colonization of the liver by CRC cells and maintenance of liver metastatic CRC tumor growth has been shown to depend on an LGR5+ tumor cell population as has been recently shown by the laboratory of Frederic Sauvage (Melo FS et al., Nature, 2017). Since the authors have demonstrated that treatment with nilotinib prevented formation and progression of already established liver metastatic nodules (Fig. 3), it would be very interesting to address whether nilotinib or inactivation of DDR1 by siRNA-mediated knock-down reduces the level of LGR5 expression in this setting.

We thank the reviewer for this insight. Unfortunately, due to time constraints, we faced technical challenges to perform IHC staining for LGR5 on experimental liver metastases. As an alternative, we analyzed the level of LGR5 transcripts by qPCR and showed that DDR1 depletion indeed leads to a downregulation of LGR5 transcripts in HCT116 cells (new Fig EV5A).

Referee #2 (Comments on Novelty/Model System):

My reservations on the medical impact are due to the fact that the authors have not clearly addressed whether their therapeutic strategy will be effective in established metastatic lesions. This is discussed in the comments to the authors.

Referee #2 (Remarks):

The metastatic dissemination of colorectal (CRC) cancer, most frequently to liver and lung, is the main cause of mortality in these patients. The manuscript by Jeitany and co-workers describes a novel strategy to diminish the invasiveness of CRC cell lines in vitro and liver metastasis following intrasplenic injection in nude mice.

In this manuscript, the authors have used several human CRC cell lines to evaluate the anti-metastatic capacity of the clinically approved TKI nilotinib. Although the rationale for selecting this drug is not clearly discussed, the authors convincingly demonstrate that the anti-metastatic capacity of nilotinib is mediated by the inhibition of DDR1. This is achieved using a combination of phosphoproteomics and several in vitro and in vivo approaches. Furthermore, the current study identified BCR as a novel DDR1 substrate. BCR has been previously reported to function as a negative regulator of beta-catenin in CRC. Jeitany and colleagues now show that DDR1-mediated phosphorylation on Tyr177 inactivates BCR resulting in increased expression of beta-catenin transcriptional targets.

One of the concerns typically associated to studies using imatinib derivatives is their broad specificity and therefore the difficulty to validate the importance of individual drug-targets. However, the authors have carried out elegant gain and loss of function experiments (in particular reconstitution experiments with the nilotinib resistant DDR1T701I mutant) to convincingly demonstrate the central role of DDR1 inhibition in mediating the nilotinib response. This was particularly important to rule out the implication of ABL as the kinase responsible for the Tyr177 phosphorylation of BCR.

Finally, the authors describe that DDR1 overexpression is associated with shorter progression-free and overall survival in CRC patients and that DDR1 activity might be increased in metastatic compared to primary lesions. As such, the authors propose that DDR1 inhibition could be effective in patients with metastatic CRC.

In sum, the research described here was excellently conceived, executed and presented. Yet I feel that additional evidence should be incorporated before the manuscript is ready for publication.

We thank the reviewer's comments, which are very positive.

Major points:

1. Central to their proposed model (see Figure 8) is the role of beta-catenin downstream of DDR1. In brief, BCR interacts with beta-catenin in the cytoplasm to prevent its nuclear localization resulting in repression of its target genes. In this model DDR1-dependent BCR Tyr177 phosphorylation disrupts this interaction thereby relieving beta-catenin inhibition. To prove this and to evaluate the impact of nilotinib and DDR1 in beta-catenin function the authors have used a beta-catenin reporter plasmid (Fig 6A, B) and have also assessed the levels of various transcriptional targets (Fig 6C). Yet, given the central role in their model they should also provide experimental evidence showing that both nilotinib treatment and DDR1 depletion increase the pool of nuclear beta-catenin. This would be particularly informative in the liver metastasis model described in figure 3F. Immunohistochemistry staining of liver sections to evaluate the subcellular localization of beta-catenin following nilotinib treatment and DDR1 gain/or loss of function experiments would reinforce their proposed hypothesis.

We have addressed the role of DDR1 expression and nilotinib treatment on the level of nuclear active β -catenin in experimental metastatic nodules described in Fig 3F. Consistent with our model, IHC analyses show that DDR1 expression significantly increases the level of active β -catenin in the nucleus of metastatic CRC cells (new Fig EV5C), while nilotinib treatment reduces this molecular response (new Fig 6F).

2. The concept of DDR1 mediating a RAS-independent mechanism in CRC is profusely used during the manuscript. Yet, in my opinion the authors have not proved that this is the case. It is true that they demonstrate that DDR1 activation (page 9) or nilotinib treatment (page 10) fails to affect MAPK or AKT phosphorylation. Yet, this experiment shows that DDR1 is not essential for the activity of these two pathways downstream of RAS, but by no means rules out the possibility that DDR1 function requires RAS and/or is somehow regulated by RAS activity. Indeed, 9 out of the 11 CRC cell lines in which the authors have assessed the effect of nilotinib inhibition on cell invasion (see Figure 1) have either KRAS or BRAF oncogenic mutations; only SW48 and Caco2 are wild-type for both. Incidentally, one of the two in which nilotinib has no effect is indeed SW48.

In order to prove the RAS independent role of DDR1 in mediating migration of CRC cells the authors would require inhibiting KRAS function and evaluate the impact on DDR1 activity. This is not an easy experiment but at least the authors could assess the effect of MEK inhibition in the context of collagen-activated DDR1. For instance, is BCR phosphorylation on Tyr177 following collagen treatment reduced in the presence of MEK inhibitors?

In the absence of this result the authors should be cautious when referring to the putative RAS-independent therapeutic strategy.

We have addressed the RAS-independent nature of DDR1 signaling using a MEK kinase inhibitor in both HCT116 and DDR1-SW620 (new Fig EV2C). We now show that MEK inhibition has no significant effect on DDR1 expression, activation and signalling (pTyr177 BCR). This notion is further supported with our primary culture of metastatic CRC cells CPP30. We now show that patient-derived CPP30 cells are WT for KRAS and BRAF (Referees Figure 3). We then used this cellular model to replicate the anti-metastatic activity of nilotinib in an additional independent KRAS/BRAF wild-type cell line. We have now *in vivo* evidence showing that nilotinib treatment significantly reduces the liver metastatic burden in nude mice that have been inoculated with these patient-derived tumour cells (new Fig 7I). Overall, this experimental data further supports the RAS-independent nature of DDR1 metastatic activity in CRC.

Referees Figure 3. KRAS and BRAF mutational status in CPP cell lines.

3. The authors suggest that nilotinib treatment might also affect the growth of established metastasis. In support of this hypothesis the authors provide the following data: "nilotinib treatment prevented liver metastatic progression, as confirmed by the significant decrease of ctDNA level compared with DMSO treated animals (Fig 3F)". While this could be true, it is also possible that in this context nilotinib might be inhibiting the release of CRC cells from the primary spleen injection from day 7 onwards (that is when nilotinib treatment starts). There is a much more direct way of assessing the effect on metastasis growth. The cell line used in this experiment to perform the intrasplenic injection is luciferase positive. Indeed, the authors only start the nilotinib treatment "when luciferase-positive metastases were already detectable" (see page 8). Being this the case, the authors could monitor metastasis growth during the course of the two-week nilotinib treatment by

also
using luciferase signal as a surrogate marker of liver tumour burden.

We have data on luciferase signal as a surrogate marker of liver tumour burden and this data has been shown in the new Appendix Fig S1.

We also have in vivo evidence showing that nilotinib treatment significantly reduces metastatic activity of KRAS WT CRC cells CPP30 (new Fig 7I). Drug treatment has been performed at day 6 post-inoculation of CRC cells in recipient mice (expectation of micrometastatic nodules in the liver of recipient animals). These important data further confirm the DDR1 role on metastatic growth and the interest of nilotinib in metastatic CRC.

Minor points:

a. Given the difficulty to access patient samples it would have been ideal to study the phosphorylation of BCR Tyr177 in the patient samples primary/metastasis shown in Figure 7B and to evaluate its correlation with DDR1 activity.

We could not get any convincing signal with anti-BCR pTyr177 antibodies from our western-blot analyses, probably because this labile phosphorylation has been subjected to dephosphorylation during tissue samples processing before storage.

Referee #3 (Comments on Novelty/Model System):

This is a solid paper which conveys novel information in the field of colorectal cancer translational research. The introduction provides extensive relevant background information (some of which can be shortened), but the objectives of the work could be better spelt out. The results are presented in a logical workflow, from descriptive data in preclinical models, to mechanistic insights, then relevance in human CRC with studies on patient samples (including in vitro experiments on patient derived models). The results are accurately described and the discussion is sound with pertinent references to previous works. The experimental procedures have been clearly reported with sufficient details to allow replication.

I believe this work could be of interest to the readers of EMBO molecular medicine, after addressing major points #1 (need to use not mutated cell models), #2 and #3 (multivariate statistical analysis should be performed) that I have outlined in the authors' comments. Experiments to address point #4 may be considered beyond the scope of this work (which is already quite extensive), but I suggest the point should at least be discussed.

Referee #3 (Remarks):

This elegant work reports that nilotinib can inhibit colorectal cancer cell invasion in vitro, and it can also reduce liver metastasis formation when RAS mutant colon cancer cells are inoculated intrasplenically in mouse models. Nilotinib impairs invasion by inhibiting collagen I dependent phosphorylation of DDR1 in CRC cells (including lines derived from patient circulating tumor cells). Mechanistically, nilotinib prevents DDR1-mediated BCR phosphorylation on Tyr177, which, in turn, is important for maintaining beta-catenin transcriptional activity necessary for cancer cell invasion. Phosphorylation of DDR1 increases over tumor progression stages, and DDR1 expression is apparently correlated with poor prognosis in advanced CRC patients.

The study is nicely presented and it describes some potentially novel and relevant information for the field of translational colorectal cancer research. However, a number of points should be clarified or extended.

We thank the reviewer's for his/her very positive comments.

Major Points

1) Most of the in vitro invasion assays and all the in vivo experiments have been performed using RAS mutated (HCT116, SW620 and CPP19) or BRAF mutated (HT29, CPP30, CTC44, CTC45) tumour cells. It is not clear whether DDR1 inhibition by nilotinib is equally effective in the context of RAS/BRAF wild-type colorectal cancer cells. For instance, nilotinib is able to reduce liver metastasis formation of HCT116 cells injected intrasplenically (Fig. 1D). This experiment should be replicated in at least an independent RAS/BRAF wild type cell line and, if possible, also using a RAS/BRAF mutant patient CTC derived cell line.

We now show that patient-derived CPP30 cells are WT for KRAS and BRAF (Referees Figure 3). This observation further supports the RAS/RAF-independent nature of DDR1 invasive activity. We then used this cellular model to replicate the anti-metastatic activity of nilotinib in an additional independent KRAS/BRAF wild-type cell-line. We have now in vivo evidence showing that nilotinib treatment significantly reduces the liver metastatic burden in nude mice that have been inoculated with these patient-derived tumour cells (new Fig 7I). However, we did not perform a similar type of experiment for CTCs due to a lack of time.

2) Does nilotinib affect viability or proliferation of CRC cells? The answer is most likely no, but data should be provided and this point should be clearly mentioned within the text.

We have now data showing that nilotinib poorly affects the standard proliferation of six different CRC cell lines tested in vitro. This piece of data is now included in the new Appendix Fig S3.

3) Expression level of DDR1 seems associated with shorter overall survival in patients with metastatic CRC. When establishing the prognostic value of a novel marker, multivariate analyses should be performed to rule out the confounding effects of any other variables known to be associated with survival. In the metastatic setting, several baseline variables are known to affect prognosis, including, RAS/BRAF mutational status, tumor location (right vs left), MSI-high status, ECOG performance status, mucinous histology, primary resection, time to metastasis (metachronous vs synchronous mets), number of metastatic sites, transcriptional subtypes. Are any of these variables associated with higher expression levels of DDR1?

A table given both univariate and multivariate analyses for several clinical characteristics has been added (Appendix Table S1). Univariate analysis demonstrates a correlation between survival (progression free survival or overall survival) and DDR1 expression, WHO performance status, tumour location, grade, number of metastatic sites, MSI, and the molecular subtype of the primary tumour (CMS). Using multivariate regression analysis, DDR1 expression remains significantly associated with the progression-free survival, together with number of metastatic sites. Some known factors like BRAF/KRAS status were not significant in our test, because a too small fraction of the patients has been analyzed (only 33/143). None of these variables are associated with DDR1 expression; in particular, there is no significant difference in the expression level of DDR1 between the four CMS subtypes (Appendix Fig S2). A comment on that has been added in p14.

4) All experiments have been performed on the cancer cell compartment, while an effect of nilotinib on tumor-tumor microenvironment interaction should also be taken into account. DDR1 is known to induce extracellular matrix remodeling. It would be interesting to learn whether and how nilotinib treatment affects how colon cancer cells interact with their tumor microenvironment. For instance, the authors could investigate whether nilotinib is able to inhibit the interaction/adhesion of HCT116 tumour cells with cell types representative of the microenvironment, such as human endothelial cells, pericytes, fibroblasts and hepatocytes in 3D co-culture systems, as well as its effect on remodeling of the extracellular matrix (are specific metalloproteinases modulated by nilotinib in CRC cells?).

As specified by the editor, we did not investigate the role of the tumour microenvironment. Nevertheless, we performed qPCR of MMP1, 3, 9 and 10 in HCT116 shCtrl or depleted for DDR1. Our preliminary results show that only MMP1 and MMP9 transcripts were detectable but weakly expressed and not modulated upon DDR1 depletion (Referees Figure 4).

Referees Figure 4. MMP1 and MMP9 mRNA levels in Ctrl and DDR1 shRNA expressing HCT116 cells.

Minor Points

5) *Nilotinib does not impair migration of SW48 or DLD1 cells, and only minimally affects CTC45. Can the authors speculate about explanations underlying this cell type specific effect?*

We have no clear explanation for this specific effect since there is no reported mutation for DDR1 in SW48 and DLD1 cell lines (<http://www.cbioportal.org>). However, DDR1 seems to be weakly activated by the collagen in these two cell lines (Fig EV1A). This suggests that the invasive capacity of these CRC cells may not be dependent of DDR1 signalling, which could explain why nilotinib does not impair cell invasion. Additionally, we could speculate a high level of β -catenin oncogenic activity in these CRC cells that may bypass the need of upstream signals emanated from DDR1.

6) *The introduction is overtly long and could be shortened by providing only the essential information about metastatic colorectal cancer (relatively) poor prognosis, and the use of TK inhibitors in CML (pages 3-4).*

The introduction has been shortened (The “leukemia” paragraph has been deleted and the introduction of ABL inhibitors has been incorporated in the 3rd paragraph).

7) *Page 6, first paragraph. It is not clear why the authors have chosen to analyze a KRAS mutant cell line (HCT116) 'To search for RAS-independent therapeutic strategies for metastatic CRC'. I would rephrase the paragraph saying that nilotinib displays anti-invasive activity in a panel of CRC cell lines, irrespective of their genotype. Indeed, Figure 1C can be improved by providing the genotype below the name of CRC cell lines in order to make it more immediate that nilotinib can reduce cell invasion independent of the tumor mutational status (if this is the case).*

This point has been modified accordingly (results section, p5).

8) *I invite the authors to speculate and discuss on possible combinations of nilotinib with agents targeting the Wnt-Beta-catenin pathway.*

A comment in the discussion has been added on potential anti-tumoral activity of the combination (p19, last paragraph).

9) *Representative pictures of cell line invasion assays used for the quantification histograms shown in Figure 1C should be provided as supplemental information.*

When this set of experiments has been conducted at the beginning of the project, the analyses were performed manually and pictures were not automatically taken. Now pictures are systematically taken and invasive cells are counted by automatic imaging methods. As a matter of time we were not in a position to defrost and repeat all these invasion assays to take new pictures.

Thank you for the submission of your revised manuscript to EMBO Molecular Medicine. We have now received the enclosed reports from the referees that were asked to re-assess it. As you will see the reviewers are now globally supportive and I am pleased to inform you that we will be able to accept your manuscript pending following a few final amendments.

1) Please address the minor text changes commented by referees 1 and 3.

***** Reviewer's comments *****

Referee #1 (Comments on Novelty/Model System for Author):

The revised version has experienced significant improvements. Overall, the chosen model systems, technical quality, and data analyses are at a high level. Although some of the concepts have been shown for other types of cancer, the treatment strategy on colorectal cancer is new. Overall, the manuscript provides very interesting data and conclusions relevant for the field of cancer translational medicine.

Referee #1 (Remarks for Author):

The authors have addressed most of the points initially under criticism by performing additional experiments which has strengthened their hypotheses and significantly has improved the quality of the manuscript. The impact of DDR1 signaling on Wnt activity, and how Nilotinib inhibits this pro-metastatic pathway in a KRAS-MAPK-independent manner, is now well characterized. Importantly, the role of BCR in this context has been analyzed in more detail. Although the functional contribution of DDR-mediated BCR phosphorylation to formation of liver metastasis *in vivo* remains enigmatic due to technical limitations, the authors have provided convincing additional data that underline the importance of phospho-BCR for accumulation of active CTNNB in the nucleus which in turn positively impacts on cell migration and invasion.

Minor point:

Since depletion of phosphorylated BCR, which is according to the authors unable to sequester CTNNB to the cytoplasm, leads to a reduced migratory capacity of cells, it might be suitable to discuss that phosphorylated BCR likely possesses an additional pro-migratory activity independent of affecting CTNNB mere localization.

Overall, the revised study by Jeitany M et al. is conducted excellent and written very clearly. By providing a novel treatment strategy for metastatic CRC and by deciphering the molecular signaling pathway affected by the drug nilotinib, this appealing study will gain a broad readership in the field of molecular and translational cancer medicine.

Referee #2 (Remarks for Author):

The authors have convincingly addressed all my concerns. In my opinion the manuscript has improved and is now ready for publication in Embo Mol Med.

Referee #3 (Remarks for Author):

The authors have satisfactorily addressed my main concern. While no more experimental work is needed, the manuscript might benefit from a few text edits/clarifications.

The novel set of results indicates that nilotinib can exert its activity on several (but not all) colorectal cancer models - irrespective of their RAS status. For this reason, I believe the abstract should have been reworded and shortened. The following sentences might be a bit out of focus for the abstract itself 'For instance, patients harbouring oncogenic mutations in RAS signalling do not respond to anti-EGFR targeted treatment. Therefore, RAS-independent therapies are needed'.

In addition, I invite the authors to acknowledge in the discussion that the migration ability of certain colorectal tumors (for instance DLD1 and SW48) is not impacted by nilotinib.

A sentence in the manuscript introduction should also be improved for clarity. Line 5 - The sentence 'survival after diagnosis is less than 5 years' does not make much sense unless placed in the appropriate context and with cited references. I guess the authors here refer to the poor 5-year-survival rates of patients diagnosed with metastatic disease, or those who relapse with metastatic CRC - but this should be clarified.

Figure 7, panel B - spelling should be 'metastatic' instead of 'metastasic' (nodules). And I guess that in the inset panel on the right 'metastase' should be edited to 'metastasis'.

2nd Revision - authors' response

10 January 2018

Referee #1 (Comments on Novelty/Model System for Author):

The revised version has experienced significant improvements. Overall, the chosen model systems, technical quality, and data analyses are at a high level. Although some of the concepts have been shown for other types of cancer, the treatment strategy on colorectal cancer is new. Overall, the manuscript provides very interesting data and conclusions relevant for the field of cancer translational medicine.

Referee #1 (Remarks for Author):

The authors have addressed most of the points initially under criticism by performing additional experiments which has strengthened their hypotheses and significantly has improved the quality of the manuscript. The impact of DDR1 signaling on Wnt activity, and how Nilotinib inhibits this pro-metastatic pathway in a KRAS-MAPK-independent manner, is now well characterized. Importantly, the role of BCR in this context has been analyzed in more detail. Although the functional contribution of DDR-mediated BCR phosphorylation to formation of liver metastasis in vivo remains enigmatic due to technical limitations, the authors have provided convincing additional data that underline the importance of phospho-BCR for accumulation of active CTNNB in the nucleus which in turn positively impacts on cell migration and invasion.

Minor point:

Since depletion of phosphorylated BCR, which is according to the authors unable to sequester CTNNB to the cytoplasm, leads to a reduced migratory capacity of cells, it might be suitable to discuss that phosphorylated BCR likely possesses an additional pro-migratory activity independent of affecting CTNNB mere localization.

Overall, the revised study by Jeitany M et al. is conducted excellent and written very clearly. By providing a novel treatment strategy for metastatic CRC and by deciphering the molecular signaling pathway affected by the drug nilotinib, this appealing study will gain a broad readership in the field of molecular and translational cancer medicine.

We thank the reviewer for his/her positive comments. We have added a comment in the discussion on the beta-catenin-independent function of BCR.

Referee #2 (Remarks for Author):

The authors have convincingly addressed all my concerns. In my opinion the manuscript has improved and is now ready for publication in Embo Mol Med.

We thank the reviewer's comments, which are very positive.

Referee #3 (Remarks for Author):

The authors have satisfactorily addressed my main concern. While no more experimental work is needed, the manuscript might benefit from a few text edits/clarifications.

The novel set of results indicates that nilotinib can exert its activity on several (but not all) colorectal cancer models - irrespective of their RAS status. For this reason, I believe the abstract should have been reworded and shortened. The following sentences might be a bit out of focus for the abstract itself 'For instance, patients harbouring oncogenic mutations in RAS signalling do not respond to anti-EGFR targeted treatment. Therefore, RAS-independent therapies are needed'.

In addition, I invite the authors to acknowledge in the discussion that the migration ability of certain colorectal tumors (for instance DLD1 and SW48) is not impacted by nilotinib.

We have shortened the abstract and added a comment on the existence of DDR1-independent CRC tumours in the 1st paragraph of the discussion.

A sentence in the manuscript introduction should also be improved for clarity. Line 5 - The sentence 'survival after diagnosis is less than 5 years' does not make much sense unless placed in the appropriate context and with cited references. I guess the authors here refer to the poor 5-year-survival rates of patients diagnosed with metastatic disease, or those who relapse with metastatic CRC - but this should be clarified.

Indeed, the sentence needed clarification. We have corrected it accordingly.

Figure 7, panel B - spelling should be 'metastatic' instead of 'metastasic' (nodules). And I guess that in the inset panel on the right 'metastase' should be edited to 'metastasis'.

We have corrected misspelling in Figure 7, panel B.

Corresponding Author Name: Serge Roche & Audrey Sirvent

Manuscript Number: EMM-2017-07918